

# Deglaciation and abrupt events in a coupled comprehensive atmosphere–ocean–ice sheet–solid earth model

Uwe Mikolajewicz[1], Marie-Luise Kapsch[1], Clemens Schannwell[1], Katharina D. Six[1], Florian A. Ziemen[1,2], Meike Bagge[3,4], Jean-Philippe Baudouin[5], Olga Erokhina[1,6], Veronika Gayler[1], Volker Klemann[3], Virna L. Meccia[1,7], Anne Mouchet[1,8], and Thomas Riddick[1]

[1]Max Planck Institute for Meteorology, Hamburg, Germany
[2]now at: Deutsches Klimarechenzentrum, Hamburg, Germany
[3]Helmholtz Centre Potsdam, German Research Centre for Geosciences - GFZ, Potsdam, Germany
[4]now at: Federal Institute for Geosciences and Natural Resources, Hannover, Germany
[5]Department of Geosciences, University of Tübingen, Tübingen, Germany
[6]now at: Astronomisches Rechen-Institut, Center for Astronomy of Heidelberg University, Heidelberg, Germany
[7]now at: National Research Council, Institute of Atmospheric Sciences and Climate, Bologna, Italy
[8]now at: GeoHydrodynamics and Environment Research, University of Liège, Liège, Belgium

**Correspondence:** Uwe Mikolajewicz (uwe.mikolajewicz@mpimet.mpg.de)

**Abstract.**

During the last 20,000 years the climate of the earth has changed from a state much colder than today with large ice sheets in North America and Northwest Eurasia to its present state. The fully-interactive simulation of this transition represents a hitherto unsolved challenge for state-of-the-art climate models. We use a novel coupled comprehensive atmosphere–ocean–vegetation– ice sheet–solid earth model to simulate the transient climate evolution from the last glacial maximum to preindustrial times. The model considers dynamical changes of the glacier mask, land–sea mask and river routing. An ensemble of transient model simulations successfully captures the main features of the last deglaciation, as depicted by proxy estimates. In addition, our model simulates a series of abrupt climate changes, which can be attributed to different drivers. Abrupt cooling events during the glacial and the first half of the deglaciation are caused by Heinrich-event like ice-sheet surges, which are part of the model generated internal variability. We show that the timing of these surges depends on the initial state and the model parameters. Abrupt events during the second half of the deglaciation are caused by a long-term shift in the sign of the Arctic freshwater budget, changes in river routing and/or the opening of ocean passages.

## 1 Introduction

The last deglaciation is marked as the transition from the cold climate of the Last Glacial Maximum (LGM; about 21 kiloyears before present, hereafter referred to as ka) to the modern climate. Driven by gradual changes of the orbital parameters and increasing atmospheric $CO_2$ concentrations, the climate warmed by several Kelvin and the massive ice sheets over North America and North Eurasia vanished, leading to a strong rise in sea level. Reconstructions of the global mean sea level show a monotonic rise (Fairbanks, 1989), although with very different rates of change. During the main phase of the deglaciation, from





16.5 to 8 ka, the average rate of 12 m ka$^{-1}$, corresponding to approx. 0.14 Sv (1 Sv = $10^6$ m$^3$s$^{-1}$) net meltwater input (Lambeck
et al., 2014; Hemming, 2004), varied between periods with greater change, e.g. during the meltwater pulse 1A (MWP1A, about
14.7 to 13.5 ka; e.g. Carlson et al., 2007; Steffensen et al., 2008; Deschamps et al., 2012; Buizert et al., 2014) with meltwater
input rates close to 0.5 Sv (Fairbanks, 1989; Deschamps et al., 2012) , and lesser change, e.g. during the Younger Dryas (YD,
about 12.8 to 11.7 ka, e.g., Buizert et al., 2014). During the glacial and the early deglaciation, several abrupt cooling events
can be detected in sediment cores from the North Atlantic containing prominent layers of ice-rafted debris, which originate
from melted icebergs that were released from the Laurentide ice sheet into the North Atlantic (Heinrich, 1988). Any addition of
large amounts of freshwater into the North Atlantic, whether through melting of icebergs or the direct release of meltwater from
the ice sheets themselves, affects the surface salinity, reduces the deepwater formation and weakens the Atlantic meridional
overturning circulation (AMOC). As a result, the northward heat transport reduces and lowers the release of ocean heat into the
atmosphere, causing large-scale climate changes. Examples of abrupt events are the cold Heinrich Stadial 1 (H1; about 16.8
ka; Hemming, 2004), the YD, or the rapid Bølling-Allerød (BA) warming, which occurred in between H1 and the YD (about
14.7 to 14.2 ka; e.g. Severinghaus and Brook, 1999; Clark et al., 2002; Weaver et al., 2003; Steffensen et al., 2008).

Various proxy data from marine sediment cores have been used to assess the AMOC state for the glacial climate (Howe
et al., 2016) and its variability (Clark et al., 2002; McManus et al., 2004; Lynch-Stieglitz et al., 2014). These data indicate
the existence of abrupt events, but the exact changes in the characteristics of the AMOC remain poorly constrained (e.g.,
Pöppelmeier et al., 2023). The common consensus, based on studies of the stable $\delta^{13}$C isotope or neodymium, is that the
position of the AMOC core was shallower during the LGM compared to modern day, but from the current set of available
proxy data, it is unclear if and how much the AMOC strength may have changed in comparison to modern state (see e.g.
review of Liu, 2023). From an ensemble of simulations with an Earth System Model of Intermediate Complexity (EMIC)
with different AMOC strengths and four proxy tracers, Pöppelmeier et al. (2023) concluded that the global proxy distributions
can only be reconciled with an ocean state characterized by a relatively weak AMOC showing a reduction in the circulation
strength by $36 \pm 8\%$ ($6.3 \pm 1.4$ Sv) relative to that of the pre-industrial state (PI). However, the sources of uncertainty in proxy
data are numerous and diverse. These include an incomplete understanding of biological processes involved in recording and
storing climate signals through proxy data (Dolman and Laepple, 2018; Liu, 2023), as well as methodological weaknesses in
the reconstruction of the water mass mixing from apparently constant signatures of water mass endmembers (e.g. Howe et al.,
2016; Oppo et al., 2018; Zhao et al., 2019; Pöppelmeier et al., 2023).

Despite the evidence of abrupt events during the deglaciation from proxies, the causes of the abrupt warming and cooling
events and their link to AMOC variations remain uncertain. To understand the AMOC variability and its drivers, therefore re-
quires modelling approaches that span the entire deglaciation. Early modelling studies focused on exploring hypotheses around
abrupt events with idealised sensitivity experiments, such as the effect of meltwater release from the ice sheets on the stability
of the AMOC (Maier-Reimer and Mikolajewicz, 1989; Schiller et al., 1997; Stouffer et al., 2006; Smith and Gregory, 2009).
More recently, the performance of CMIP-style (Coupled Model Intercomparison Project) climate models and their sensitivities
to climate perturbations have been evaluated in time slice experiments of, for example, the LGM or PI (e.g. Harrison et al.,
2015; Kageyama et al., 2021). Historically, these CMIP-style climate models have been developed for the simulation of time





periods in which changes in the topography and the ice sheets can be neglected. This assumption, does not hold for the last
deglaciation and, together with computational demands, represents one of the technical challenges that in large parts explains
the paucity of transient simulations covering the last deglaciation with CMIP-style climate models. Furthermore, transient
deglaciation simulations have so far relied on prescribed ice sheets from reconstructions (Liu et al., 2009; Obase and Abe-
Ouchi, 2019; He et al., 2021; Kapsch et al., 2022; Bouttes et al., 2023; Obase et al., 2023). Several studies have highlighted
that the transient climate response in such simulations, particularly around abrupt events, is dominated by the choice of ice
sheet reconstruction and their type of meltwater forcing (Kapsch et al., 2022; Bouttes et al., 2023). A primary example is the
meltwater release associated with MWP1A. The main ice sheet reconstructions utilised in PMIP4 (Tarasov et al., 2012; Peltier
et al., 2015; Ivanovic et al., 2016) suggest strong meltwater forcing in excess of 0.4 Sv in the northern hemisphere for MWP1A.
The meltwater release of MWP1A into the North Atlantic results in an AMOC weakening, and cold surface temperatures in
the northern hemisphere. This is not consistent with the subsequent BA warm period as recorded in the Greenland temperature
records (Kapsch et al., 2022). This discrepancy between the prescribed meltwater forcing from ice-sheet reconstruction and
the ensuing climate response advocates for the inclusion of interactive ice sheets into climate models. However, because of
the long characteristic time scales involved in the modelling of ice sheets, they have been mainly included in EMICS (Charbit
et al., 2005; Roche et al., 2014; Heinemann et al., 2014; Ganopolski and Brovkin, 2017). The reduced physics that EMICs
are based on permit the longer integration times necessary for these simulations. Coupling of interactive ice sheets into more
computationally expensive CMIP-style climate models has so far focused mainly on shorter time scales up to two millennia
(e.g. Ridley et al., 2005; Mikolajewicz et al., 2007a, b; Vizcaíno et al., 2008; Ziemen et al., 2014; Vizcaíno et al., 2015; Munt-
jewerf et al., 2020). However, previous modelling efforts often included only a subset of the ice sheets (Charbit et al., 2005;
Muntjewerf et al., 2020; Quiquet et al., 2021), employed coupling routines in which the more expensive climate model was
accelerated (Charbit et al., 2005; Heinemann et al., 2014; Ziemen et al., 2019) and/or did not explicitly account for iceberg or
solid earth interactions.

Here we present results from a novel model system that consists of a coarse resolution version of a CMIP-type climate
model, includes interactive ice sheets for both hemispheres as well as iceberg and solid earth interactions, and is applied to
the last deglaciation. The model system is described in section 2 and its performance is evaluated for the LGM and PI time
slices (Section 3.1). We then present an ensemble of transient simulations of the last deglaciation with the new model sys-
tem (Section 3.2) before we focus on the consequences and control of abrupt climate events that are triggered by ice-sheet and
topography changes and the associated climate response (Sections 4.1 and 4.2). The main findings are summarised in section 5.





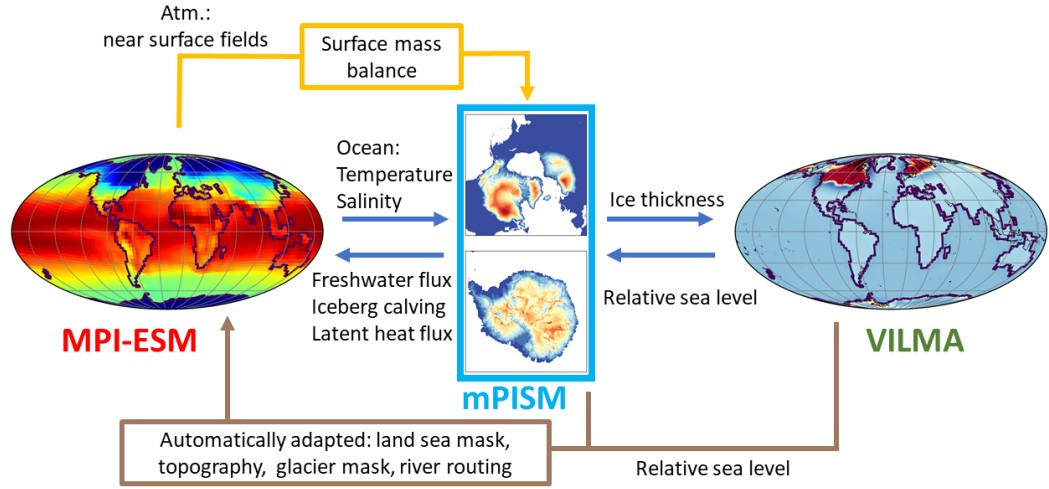

**Figure 1.** Schematic of the model framework MPI-ESM/mPISM/VILMA with its individual components MPI-ESM, mPISM, and VILMA and the information exchange across the components. As an example, one characteristic property is shown for each model component: surface temperature for MPI-ESM, ice-sheet thickness for mPISM, and relative sea level change for VILMA.

## 2 Model and experiment description

### 2.1 Model

The model system MPI-ESM/mPISM/VILMA consists of a coupled atmosphere–ocean-vegetation–ice sheet–solid earth model. A schematic of the coupled model system is shown in Fig. 1. We provide here only a very brief description of the individual components to focus on the scientific applications.

The climate model is the coarse resolution version of Max Planck Institute for Meteorology Earth System Model (MPI-ESM) version 1.2 (Mikolajewicz et al., 2018; Mauritsen et al., 2019) that also has been used for the deglacial simulation described in Kapsch et al. (2022). The atmospheric component is ECHAM6.3 (Stevens et al., 2013). It has a spectral resolution of T31 (corresponding to approx. 3.75° on a Gaussian grid) and 31 levels in the vertical on hybrid coordinates. Land surface processes and vegetation dynamics on land are treated by the JSBACH component (Reick et al., 2013). A hydrological discharge model (Hagemann and Dümenil, 1998) transports the runoff from land to the ocean.

The ocean component is MPIOM1.6 (Jungclaus et al., 2013). It has a resolution of 3° on an Arakawa C (Arakawa and Lamb, 1977) with the grid poles located in southeast Greenland and in the centre of Antarctica, yielding a relative high resolution in the North Atlantic subpolar gyre (35 to 150 km). The ocean model has 40 levels on z-coordinates, with layer thickness increasing towards depth, a free surface and a mass flux boundary condition. Vertical mixing and diffusion are based on a



Richardson-number dependent formulation by Pacanowski and Philander (1981). In addition, a simple mixed-layer scheme
describes near-surface mixing. For the background mixing, a fixed vertical profile is assumed. Starting with a given surface
value, it increases linearly with depth until 1000 m. Below it is constant. A sea-ice component with viscous plastic rheology
following Hibler (1979) with 0-layer thermodynamics including snow is embedded in MPIOM. In addition, the version of
MPIOM used here contains a new thermodynamic Eulerian iceberg model with 5 size classes (Erokhina and Mikolajewicz,
2024), allowing the explicit treatment of calving icebergs and a 3-dimensional input of meltwater and latent heat release.

Ice sheets are represented by the thermomechanically-coupled ice-sheet model mPISM (Ziemen et al., 2019) based on
PISM0.7.3. The ice dynamics in mPISM are based on the superposition of the shallow-ice approximation (SIA) and shallow-
shelf approximation (SSA), making it suitable to model slow ice flow, fast ice flow as well as grounding-line migration.
The polar stereographic grid has a resolution of 10 km in the northern hemisphere and 15 km for Antarctica. The surface
mass balance of the ice sheet is calculated by an energy balance model (EBM) using hourly atmospheric input data from the
atmospheric model component ECHAM6.3 (Kapsch et al., 2021).

The response of the solid Earth and changes in relative sea level, induced by the consistent redistribution of ice and water
masses are computed with the global VIscoelastic Lithosphere and MAntle model (VILMA, Martinec et al., 2018). VILMA is
employed in its 1D configuration, which assumes that the viscosity structure of the Earth varies only in radial direction. VILMA
solves the sea-level equation, accounts for rotational feedback and considers an incompressible self-gravitating viscoelastic
material with a Maxwell rheology. The sea level equation is solved on the horizontally discretized Gaussian F128 grid (approx.
$0.7°$), which represents the resolution at which data are exchanged, i.e. the ice sheet distribution as input and the negative
relative sea level change, representing the change in topography due to glacial isostatic adjustment (e.g., Konrad et al., 2015).

The coupling time step between the atmosphere and ocean model is one day. The iceberg model is part of the ocean model
and thus interacts on every ocean time step. The ESM is coupled every 10 years with the ice sheet and solid earth models.
At this interval, new high resolution topographies ($1/6°$ horizontal resolution) are calculated from the results of VILMA and
mPISM and used to derive adapted topographies (incl. land/sea masks) for the ocean and the atmosphere using the algorithm
developed by Meccia and Mikolajewicz (2018). River routing directions are adapted dynamically according to topography
changes using the algorithm developed by Riddick et al. (2018). The ice sheet model supplies calving fluxes to the iceberg
module. A reduced version of this model system without icebergs, ice sheet and solid Earth dynamics has been used by Kapsch
et al. (2022) to perform deglacial simulations using prescribed ice sheet reconstructions.

## 2.2 Experiments

For all simulations, the atmospheric concentrations of $CO_2$, $CH_4$ and $N_2O$ were prescribed according to Köhler et al. (2017).
Earth orbital parameters were calculated following Berger and Loutre (1991). In addition, in most simulations, annually varying
volcanic forcing (Schindlbeck-Belo et al., 2024) was prescribed for the time after 26 ka.

To achieve a physically-consistent initial state for 26 ka, transient asynchronous spin-up simulations were started from one
existing glacial state of the model system. These simulations cover the period from 45 to 26 ka. While atmosphere, ocean and





vegetation were integrated for only 10 years to save computing time, the ice sheet and solid earth components were integrated for 100 yr with the forcing calculated from the last climate model run. After completion of the ice sheet and solid earth model

components, the topography was updated and the climate model was again integrated using the forcing derived from the ice sheet model. This acceleration saves considerable amounts of computing time and serves to spin-up the model components with the longest memory.

A set of eight synchronously coupled transient deglacial experiments with different model parameter settings were started from asynchronous spin-up simulations at 26 ka. For an overview of the ensemble members see Table 1.

To avoid the effect of model drift, we disregard the first 1000 years of the synchronous simulations. Thus, our analysis covers only the time from 25 ka to 1850 CE.

Effectively, our full ensemble consists of two smaller, separate sub-ensembles, each containing four members. The first sub-ensemble investigates the effect of vertical mixing in the ocean. Here, D1.1 serves as the baseline experiment. All four members of this sub-ensemble start from the same asynchronous spin-up simulation. In experiment D1.2 the vertical back-

ground mixing coefficient was increased while in experiment D1.3 it was decreased. All experiments of this sub-ensemble used a climatological volcanic forcing representing PI with the exception of D1.4, which was forced with the time-varying volcanic forcing (Schindlbeck-Belo et al., 2024).

For the second sub-ensemble, D2.1 is the baseline experiment. In comparison to D1.1, several parameter values within the individual ESM components were changed. This includes changes in sea-ice parameters towards a more rigid sea ice and

reduced friction between the ocean and sea ice. Additionally, the prescribed vertical profile of oceanic background mixing was reduced in the upper ocean. As the treatment of the ocean surface velocities in the coupling interface of the atmosphere was discovered to be affected by a bug, this part of the coupling was suppressed in all simulations of this sub-ensemble. To maintain a similar climate in terms of the surface temperatures, small changes in the atmospheric parameters were required to reduce a slight cold bias. Changes in the ice-sheet parameters in comparison to the first sub-ensemble focused on enhanced

sliding and a more sensitive basal mass balance. The goal behind the enhanced sliding tuning was two-fold: first, we targeted a faster expansion of the ice sheets towards the LGM; second, the resulting thinner ice sheets facilitate ice sheet retreat during the deglaciation. The more sensitive basal mass balance was targeted towards a faster retreat of the marine portions of the Antarctic ice sheets. All runs in this sub-ensemble were run with time-varying volcanic forcing and start from their own matching asynchronous spin-up.

In addition, two sensitivity experiments (D2.1.1 and D2.1.2) were performed to investigate the effect of initial conditions for the ice sheets on the deglacial climate trajectory. Except for the initial conditions they are identical to D2.1. These experiments are analyzed in Section 4.2 and are not included in our main ensemble. Experiment D2.1.1 was only integrated until 7 ka.

## 3   The simulated climate

To evaluate the simulated climate, we first show the model results for the time of the LGM. The LGM has been intensively

investigated for steady state time-slice simulations (21 ka), as it is a standard time slice within the Paleo Modeling Intercompar-





**Table 1.** Ensemble members with their parameter settings and spin-up procedures. Detailed individual parameter changes are listed in Table A1 and A2.

| Name | MPI-ESM | Volcanoes | PISM_ANT | PISM_NH | Parent run |
|---|---|---|---|---|---|
| D1.1 | version 1 | fixed | version 1 | version 1 | D1.1asy |
| D1.2 | version 1 <br> - increased background diffusivity in the ocean | fixed | version 1 | version 1 | D1.1asy |
| D1.3 | version 1 <br> - decreased background diffusivity in the ocean | fixed | version 1 | version 1 | D1.1asy |
| D1.4 | version 1 | varying | version 1 | version 1 | D1.1asy |
| D2.1 | version 2 <br> - reduced background mixing in the upper ocean <br> - more rigid sea ice <br> - atmosphere retuned | varying | version 2 <br> - enhanced sliding near grounding line <br> - increased basal melt | version 2 <br> - enhanced sliding <br> - increased basal melt | D2.1asy |
| D2.2 | version 2 <br> - atmosphere as in version 1 | varying | version 2 | version 2 | D2.2asy |
| D2.3 | version 2 <br> - brighter albedos in smb-EBM | varying | version 2 | version 2 | D2.3asy |
| D2.4 | version 2 | varying | version 1 <br> - decreased sliding near grounding line | version 2 | D2.4asy |
| D2.1.1 [a,b] | version 2 | varying | version 2 | version 2 | D2.1asy <br> ice sheets 27 ka |
| D2.1.2 [a] | version 2 | varying | version 2 | version 2 | D2.1asy <br> ice sheets 28 ka |

[a] The sensitivity studies D2.1.1 and D2.1.2 are not part of the ensemble, [b] D.2.1.1 was run only until 7 ka

ison Project Phase 4 (PMIP4) (e.g. Kageyama et al., 2020). In subsection 3.2 we will present the simulated transient deglacial climate changes.

## 3.1 LGM climate relative to PI

Since our model runs show substantial millennial-scale variability throughout the glacial (see Section 4.2), we chose to average
the LGM state over the time window from 23 ka to 18 ka. In the Holocene, millennial scale variability is rather small, thus





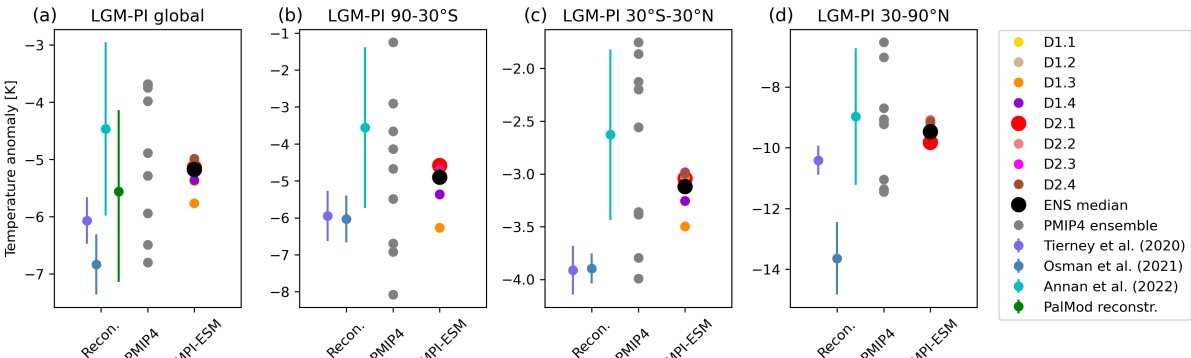

**Figure 2.** Mean annual surface air-temperature anomalies in K with respect to PI for different regions: (**a**) the global mean, the mean over (**b**) the southern extratropics (90°-30° S), (**c**) the tropics (30° S-30° N) and (**d**) the northern extratropics (30°-90° N). Each panel shows values derived from reconstructions (left) (Friedrich et al., 2016; Bereiter et al., 2018; Tierney et al., 2020; Osman et al., 2021; Annan et al., 2022, and the PalMod reconstructions), the PMIP4 model ensemble (center Kageyama et al., 2020) and our model ensemble (right). Whiskers on the reconstructions indicate uncertainties.

only 1000 years (1100 to 100 yrBP) were used for the calculation of the PI climate state. In addition to the ensemble median, minimum and maximum, we also show results from one particular experiment (D2.1), which we use in Section 4.1 to discuss the deglacial climate variability in more detail.

### *Near-surface air temperatures*

The simulated global mean LGM near-surface air temperature (SAT) in our model ensemble is 5.0 to 5.8 K colder than the PI reference state (Fig. 2 a). The simulated cooling pattern shows a strong polar amplification illustrated by the significantly larger cooling in the high northern latitudes (9.1 to 9.8 K) than in the tropics (between 3.0 and 3.5 K; Fig. 2d and c). In the southern high latitudes, the simulated LGM climate is between 4.6 and 6.3 K (mean: 5.1; median: 4.8) colder (Fig. 2b). These values lie well within the LGM (21 ka) climate signal from the PMIP4 ensembles (3.7 to 6.8 K). Our estimates also agree

well with estimates of LGM near-surface air temperature anomalies from available reanalysis/reconstruction data sets (see Fig. 2). For validation, we use recent products either based entirely on proxy data (such as the PalMod reconstructions, based on the methodology and proxy data described in Baudouin et al., 2024, see Appendix A1) or data assimilation products that use either a single model (Tierney et al., 2020; Osman et al., 2021) or an ensemble of models (Annan et al., 2022). Globally, our simulated SAT anomalies (mean: -5.2 K; median: -5.1 K) align with the range of reconstructed SAT anomalies of the PalMod

reconstructions (Appendix A1) of -4.1 and -7.1 K (mean: -5.6 K) and the reanalysis of Friedrich et al. (2016) of -6.5 to -3.5 K (mean: -5.0 K) and Annan et al. (2022) of -2.9 to -6.0 K (mean: 4.5 K). The reanalyses of Osman et al. (2021) and Tierney et al. (2020) and to a smaller extent the reconstructions of Bereiter et al. (2018) are consistently colder than our ensemble (-6.3 to -7.4 K, -5.7 to -6.5 K, and -7.5 to -5.1 K, respectively) and the Annan et al. (2022) data assimilation product. The same





signal is present through regional estimates for the northern and southern extratropics and tropics, where our ensemble mean

is slightly colder than Annan et al. (2022) but warmer than Osman et al. (2021) and Tierney et al. (2020). Thereby nearly all
ensemble members fall into the uncertainty range of the Annan et al. (2022) estimates, except for D1.3 for the tropics and
southern extratropics.

Maximal cooling is simulated over the Laurentide and Fennoscandian ice sheets (Fig. 3), where the higher surface elevation
leads to an additional cooling effect. Also the North Atlantic shows a rather strong cooling, which is related to extended sea-ice

cover (Fig. 3). In general, the simulated pattern of cooling is rather similar to the ensemble mean pattern from the CMIP6
LGM time-slice simulations (Kageyama et al., 2020, their Fig. 1) and the reanalysis product of Annan et al. (2022). The spatial
pattern of the reanalysis product by Osman et al. (2021) is also very similar to our ensemble, but the amplitude of the cooling
is considerably stronger.

*Sea-ice extent*

The simulated sea-ice extent for the PI period is fairly close to the extent derived from the HadiSST data set for the years 1900
to 1955 (Hurrell et al., 2008) (see isolines in Fig. 3). The main discrepancy is an underestimation of the sea-ice extent in the
Southern Ocean.

For the LGM, the ensemble spread for the winter sea-ice extent in the North Atlantic is considerable. Some simulations show
an ice-free Northeast Atlantic, others a state with winter sea ice that covers nearly the entire North Atlantic north of 40° N.

The Northwest Atlantic is seasonally sea ice covered in all simulations. In the North Atlantic, the GLOMAP reconstruction
(Paul et al., 2021) does not show a large difference between the Northwest and Northeast Atlantic. It is worth noting that
the simulation D2.1 does not show such a contrast either. The Norwegian Sea is only seasonally covered by sea ice in all
simulations, whereas most of the Greenland and Icelandic Sea are permanently covered by sea ice. GLOMAP indicates seasonal
sea-ice cover in the entire Nordic Seas, with the LGM summer sea-ice margin even being north of its position at the beginning

of the 20th century in the HadiSST data set (Hurrell et al., 2008). The simulated Southern Ocean sea-ice cover is seasonal,
except for some regions close to the coast of Antarctica. The model generally underestimates the summer sea-ice extent in the
southern hemisphere, both for PI as well as for LGM. The simulated winter sea-ice extent in the Southern Ocean matches the
GLOMAP data set well.

*AMOC*

The maximum strength of the North Atlantic Deep Water (NADW) cell at 26.5° N, which defines the AMOC strength, varies
in our PI ensemble between 18.3 and 24.8 Sv (ensemble median is 20.0 Sv) and is located at a depth slightly deeper than 1000
m (see profile of the AMOC at 26.5° N in Fig. 4a). From measurements of the RAPID array at the same latitude for the years
2004 to 2017, the strength of the AMOC has been estimated to be $17.0 \pm 4.4$ Sv (Frajka-Williams et al., 2019; Moat et al.,
2023). The strength of the Antarctic Bottom Water (AABW) cell at 26.5° N varies between 0.4 and 2.5 Sv for the PI ensemble.

220    For the LGM, the spread of the AMOC strength is much larger, showing values between 14.7 and 26.3 Sv, with an ensemble
median of 16.8 Sv. Except for one simulation (D1.2), all runs show a shallower and slightly weaker AMOC cell during the

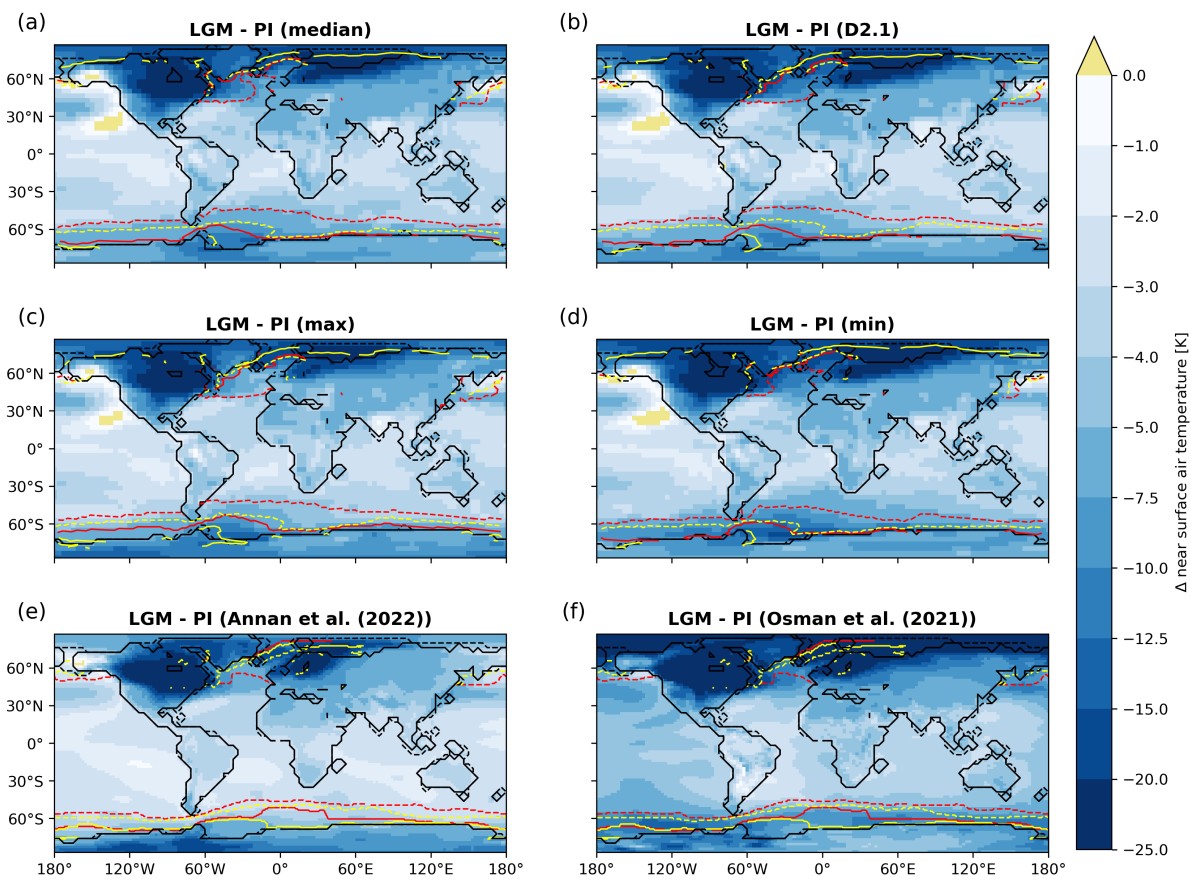

**Figure 3.** Difference in annual mean near-surface air temperature between LGM and PI in K (colours). The panels show (a) the ensemble median, (b) the results for simulation D2.1, (c) the ensemble maximum, and (d) the ensemble minimum. The isolines show sea-ice extent (> 0.15 sea-ice coverage in the long-term mean seasonal climatology) for PI summer (yellow solid) and winter (yellow dashed) and LGM summer (red solid) and winter (red dashed). The bottom panel shows reanalysis results from (e) Annan et al. (2022) and (f) Osman et al. (2021). The LGM sea-ice margin plotted in the bottom panels is taken from the GLOMAP (Paul et al., 2021) reconstruction, the PI reference has been calculated from years 1900 to 1955 from the HadiSST data set (Hurrell et al., 2008). PI and LGM land–sea masks are indicated by the solid and dashed black lines, respectively.





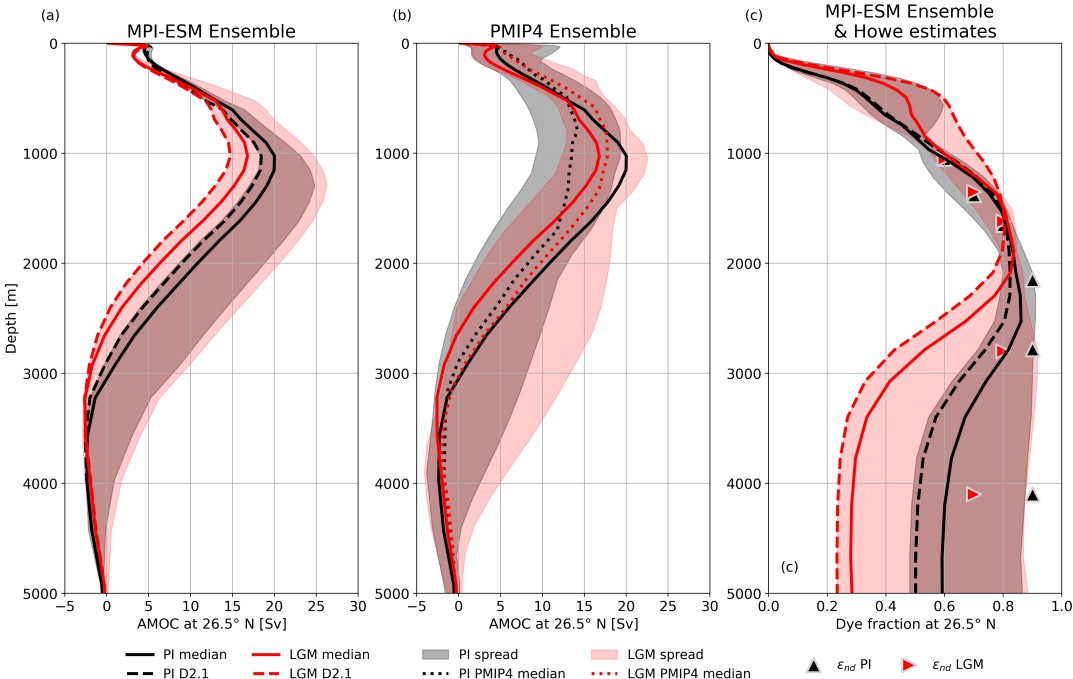

**Figure 4.** (a) Depth profiles of the simulated AMOC at 26.5° N for the ensemble median (solid lines) and the ensemble spread (shadings) for PI (black) and LGM (red). Dashed lines highlight the results of D2.1. (b) Same as (a), but based on the PMIP4 ensemble (Kageyama et al., 2020) with the ensemble median given as dotted line. For better comparison, the ensemble medians from (a) is shown as well (solid lines). (c) Depth profiles of the zonally averaged relative contribution of water originating from the surface of the North Atlantic or the Arctic at 26.5° N (dye fraction). Same color coding as in (a). Overlaid are estimates based on neodymium composition $\varepsilon_{Nd}$ from Howe et al. (2016) (black triangle for PI, red triangle for LGM).

LGM than for their PI counterpart. The vertical extent of the NADW cell, for which the lower boundary is defined as the depth where the AMOC strength equals zero below 2000 m, becomes shallower by 250 m to 450 m for almost all individual ensemble members. However, both the spread of the maximum strength of the NADW cell as well as that of its lower margin

225  are about twice as large for the LGM as compared to the PI simulations. The prescribed vertical background mixing in the ocean has a very large effect on the AMOC profile as can be seen from comparing simulations D1.2, D1.1 and D1.3 (see Fig. A1a). Reduced vertical mixing results in a prolonged preservation of the density excess of AABW compared to NADW on its path towards the deep North Atlantic. The consequence is a weakening and shallowing of the NADW cell and an enhanced presence of AABW. The effect is larger for the LGM than for PI. Thus, lower vertical mixing enhances the AMOC differences

230  between LGM and PI. Stronger vertical mixing leads to a LGM AMOC that is even stronger and deeper than its PI counterpart.

For the PMIP4 ensemble, most LGM simulations show a weak deepening and a substantial strengthening of the AMOC (Fig. 4b). This is in contrast to estimates based on proxy data that indicate a shallowing of the AMOC core and a reduction of the circulation strength (e.g Pöppelmeier et al., 2023), in line with our results.





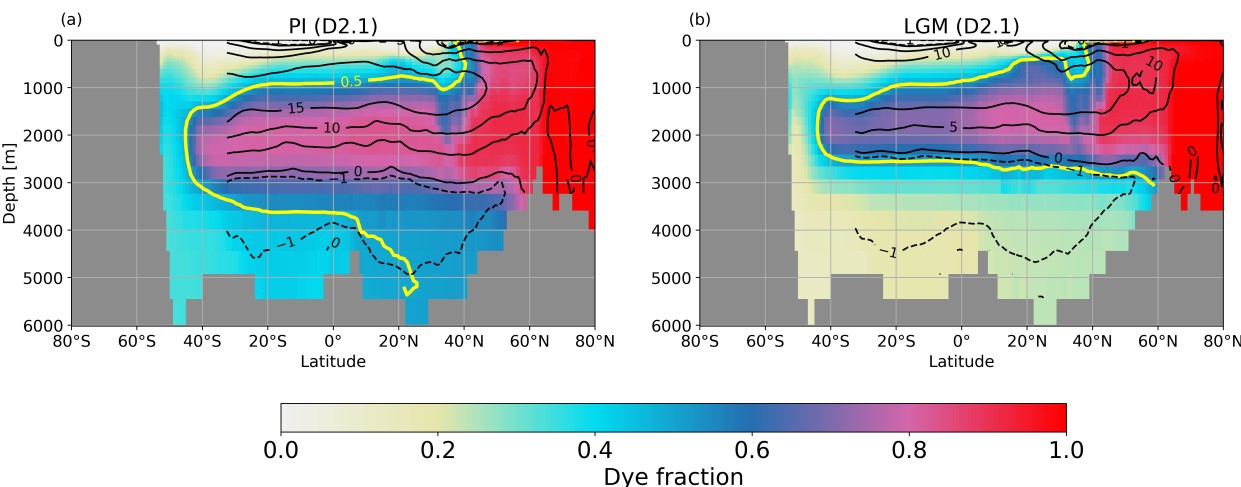

**Figure 5.** Atlantic section of the zonal mean dye fraction for (a) PI and (b) LGM for one ensemble member (D2.1). Overlaid are contour lines of the AMOC strength (black lines at -1, 0, 5, 10, 15 Sv) and one contour line of the dye fraction (yellow line = 0.5 as indicator of the boundary between NADW and AABW)

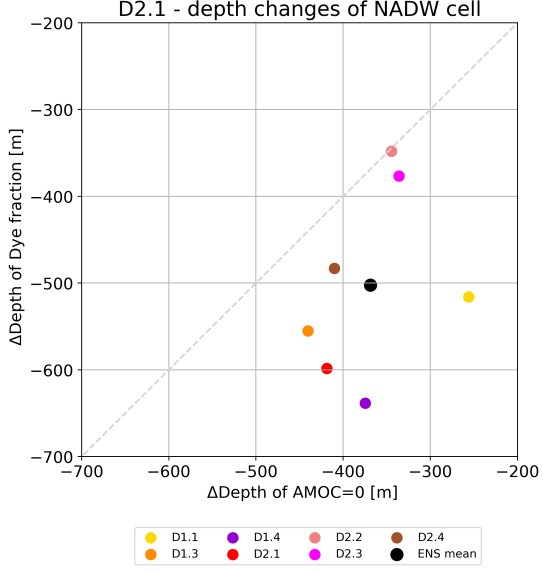

**Figure 6.** Depth changes between LGM and PI of the dye fraction for the concentration being found for AMOC=0 in PI versus that of AMOC=0 at 26.5° N. Ensemble members and ensemble mean are shown (see legend). D1.2 has been excluded from this analysis (see text). Light-grey dashed line indicates the 1:1 depth change for orientation.





For the assessment of the past circulation changes, proxy data are the only source of information. To gain a better insight into how circulation changes would be reflected in proxy data, we included an artificial water mass tracer (dye) in our simulations, which indicates the relative contribution of water originating from the surface of the North Atlantic and the Arctic, and thus is tracing the relative contribution of NADW. The dye is set continuously to 1 at the surface in the source area and to 0 outside this domain. In the ocean interior, the dye is advected and mixed like salinity. A decreasing dye fraction reflects mixing of water which originates from other source regions, which is primarily AABW in the Atlantic below 2000 m. The dye fraction at 26.5° N shows the core of NADW for PI at depths between 1500 and 3000 m (Fig. 4c). The depth of 3000 m corresponds to the lower margin of the AMOC cell. Below 3500 m, the intrusion of AABW becomes visible by the thick layer of water that consists of a mix of NADW and AABW. For the LGM, the core of the NADW lies shallower and has a reduced vertical extent. The deep layers are dominated by AABW. Estimated changes of the NADW distribution from proxy data show a qualitatively similar pattern (Howe et al., 2016; Oppo et al., 2018; Gu et al., 2020). Note, that the reconstruction of the data by Howe et al. (2016) is based on neodymium isotopes, which might underestimate changes of NADW at the LGM due to their source being at the continental slope (Liu, 2023), unlike other proxies and our dye tracer, which have their source solely at the surface. Reduced vertical mixing leads to stronger differences in the dye profile at 26.5° N between LGM and PI (see Fig. A1b) in line with the AMOC changes.

Figure 5 illustrates the distribution of the dye fraction and the location and strength of the cells of NADW and AABW at the PI and the LGM for one ensemble member (D2.1). In agreement with the shallowing of the lower boundary of the NADW cell at the LGM, the vertical extension of the dye fraction higher than 0.5 is limited to the upper 2500-3000 m. Qualitatively, the shift of the distribution of water masses between PI and LGM in our ensemble follows the changes derived from proxy data (Howe et al., 2016; Menviel et al., 2017; Oppo et al., 2018; Gu et al., 2020). However, quantitatively, the change of the AMOC geometry is different from the change in the spatial pattern of the dye fraction. If the lower boundary of the NADW cell is confined by AMOC=0, a dye fraction of about 0.5 is found at a very similar depth at the LGM. For the PI, the dye fraction at AMOC=0 is much higher (around 0.7) and NADW is the dominant water mass in the deep Atlantic north of 20° N. Our dye fraction provides the opportunity to compare simulated depth shifts of the lower boundary of the AMOC cell to changes in the location of NADW identified by the dye fraction (Fig. 6). As already apparent from D2.1 in Fig. 5, almost all ensemble members show that the shallowing of the dye fraction for the LGM is more pronounced than the shallowing of the NADW cell. Ensemble member D1.2, in which the AABW cell does not extend to 26.5° N, has been excluded from this analysis. For the ensemble mean, the depth change of the water mass boundary indicated by the dye fraction is about 35 % larger than that for the AMOC cell boundary. Note that the dye fraction has no methodological uncertainty and no potential variations in the source function as it is the case for proxy data (Howe et al., 2016; Gu et al., 2020; Liu, 2023). The simulated temporal and spatial variations of the dye fraction at AMOC=0 are a result of water mass mixing intensity and complicate the straightforward assessment of the AMOC geometry from proxy data (see a further discussion in Appendix A3 and Fig. A2). The simulated difference should therefore be taken as a cautionary note for a quantitative validation of AMOC changes with proxy-derived changes in water mass boundaries.





**Figure 7.** Ice thickness maps for the northern hemisphere and Antarctica. The column title indicates the respective time slice. The rows correspond to (a-d) the ensemble median, (e-h) D2.1, (i-l) the ensemble minimum, (m-p) the ensemble maximum, (q-t) ICE-6G, and (u-x) GLAC-1D. The thick black line indicates the median position of the grounding line.



*Ice sheets*

The simulated Laurentide ice sheet for the LGM has a similar size as in the GLAC-1D reconstructions (Fig. 7, Tarasov et al.,
2012; Briggs et al., 2014) and ICE-6G (Peltier et al., 2015), as shown in Fig. 7. The largest differences occur at the south-
eastern margin, where the reconstructions extend further south than the Laurentide ice sheet in the simulations. Moreover, the
simulated Laurentide ice sheet shows a clear two-dome structure - one dome covering the Cordilleran (western Laurentide) ice
sheet and the second, more prominent dome, covering the eastern Laurentide. The latter is not distinctly visible in the recon-
structions. However, the reconstructions exhibit considerable uncertainties, especially in the thickness distribution. Differences
in ice thickness between the modelled Laurentide ice sheet and reconstructions are most prominent in the Hudson Bay and
Mackenzie areas. Both regions frequently exhibit Heinrich-event (HE) like surge events (e.g. Heinrich, 1988; Ziemen et al.,
2019; Schannwell et al., 2023) in our model, which will be discussed in more detail in Section 4.1. The impression that the
Labrador Sea is covered by an ice shelf in our simulations stems from the fact that we average over periods with HE-like surge
events, during which a temporary ice shelf forms (Fig. 7a, e, and m), and more frequent periods without an ice shelf in the
central Labrador Sea. The model simulates a Fennoscandian ice sheet that is in good agreement with the reconstructions with
differences restricted to the eastern part of the Fennoscandian ice sheet, where it extends further east in the simulations. All
model realisations show an ice mass over Northeast Siberia that is not present in the reconstructions. For the PI, our simula-
tions show a realistic Greenland ice sheet and some smaller ice caps over Svalbard, Novaya Zemlya and other Arctic islands. In
general, the size of the ice caps over Norway, Baffin Island and the Rocky Mountains is overestimated. The largest discrepancy
to observations occurs in Northeast Siberia, where a considerable ice cap survives the deglaciation, containing between 2.4
–3.2 m sea-level equivalent.

   The comparison of simulated ice-sheet size with ice-sheet reconstructions shows larger discrepancies for Antarctica than for
the northern hemisphere. Particularly, the marine-based regions of the Filchner-Ronne and Ross ice shelves, where the ice sheet
retreated significantly during the deglaciation, are different. Similar results were obtained by previous glacial cycle simulations
with stand-alone ice sheet models (Maris et al., 2014; Pollard et al., 2017; Albrecht et al., 2020). All model realisations tend to
follow one of the following two scenarios for Antarctica: (i) the ice-sheet advances to the continental shelf edge at the LGM,
but does not retreat far enough during the deglaciation; or (ii) the ice-sheet does not advance to the continental shelf edge at
the LGM, but retreats to a position in agreement with present-day observations. This underlines the challenge of successfully
simulating the evolution of marine terminating ice sheets over the course of the last deglaciation (Albrecht et al., 2023)

## 3.2  Climate evolution over the last deglaciation

In this section, we turn our focus towards the transient climate evolution of the last deglaciation in our model ensemble. We
focus on a selection of variables for which proxy data and reconstructions are available and additional variables that give a
better physical understanding of the proposed processes.





*Near-surface air temperature*

Our simulated SATs show an almost continuous warming (Fig. 8a) akin to the trajectory of the PalMod reconstructions and those of Osman et al. (2021). Overall, the simulated warming closely follows the prescribed increase in atmospheric greenhouse gas concentrations with the strongest warming between 17 ka and 10 ka. During the Holocene, our simulations indicate a steady warming trend, which is consistent with the data assimilation product by Osman et al. (2021), but in contrast to the PalMod reconstructions, which indicate a slight cooling trend throughout the Holocene. This discrepancy in the Holocene temperature trend is quite typical. For a detailed discussion of the Holocene temperature conundrum see e.g. Kaufman and Broadman (2023).

*Sea-surface temperature*

Superimposed on the overall warming trend in our simulations are abrupt millennial- and centennial-scale cooling events (Fig. 8f, Fig. 9). This variability is not evident in the proxy based products due to the lower temporal resolution of the reconstructions incurred by the methodological design and the underlying quality of the proxy data.

The latitudinal distribution of the deglacial ocean surface warming signal is shown in Fig. 9, by comparing the sea-surface temperatures (SSTs) over different latitude bands derived from our ensemble to proxy based estimates. Similar to the global SAT evolution, the ocean surface temperature evolution in the northern extratropics shows a continuous warming throughout the entire simulation, interrupted by several abrupt cooling events. Thereby, the model spread of the transient signal is about 1 K, if the abrupt cooling events are excluded (see Fig. 9). A slow warming trend is present at all latitudes until 17 ka, after which the warming significantly accelerates. In contrast, both proxy based data sets show a temperature minimum around 17 to 16 ka, at the time of H1, followed by rapid deglacial warming which tapers off between 14 and 13 ka.

In the proxy data set, the Younger Dryas cold event peaks between 13 and 12 ka. A similar cooling signal is also modelled in some of the runs (e.g. D1.1, D1.2 and D1.4) albeit with a weaker amplitude than in the reconstructions. In general, the timing of the occurrence of individual abrupt events, however, varies between the model runs. From 12 ka onward, all curves show a strong warming until the beginning of the Holocene. Hereby, the modelled warming rates are lower than suggested by the proxy data. During the Holocene, the model runs indicate a weak warming trend, in accordance with the Osman et al. (2021) estimate.

In the tropics, the model simulations show a glacial cooling amplitude that is sandwiched by the proxy data, whereupon the PalMod reconstructions indicate less cold SSTs and the reanalysis of Osman et al. (2021) indicates cooler SSTs. The onset of the deglacial warming is overall delayed by approximately 1 kyr. The simulated warming signal here is much smoother and the millennial-scale variability seen in the northern hemisphere extratropics is much reduced. While this is in accordance with the reconstructions from PalMod, the Osman et al. (2021) reanalysis shows a temporal variability rather similar to the northern extratropics. In addition, during the early phase of the deglacial warming, the model ensemble is slightly colder than the uncertainty range from the PalMod reconstructions.





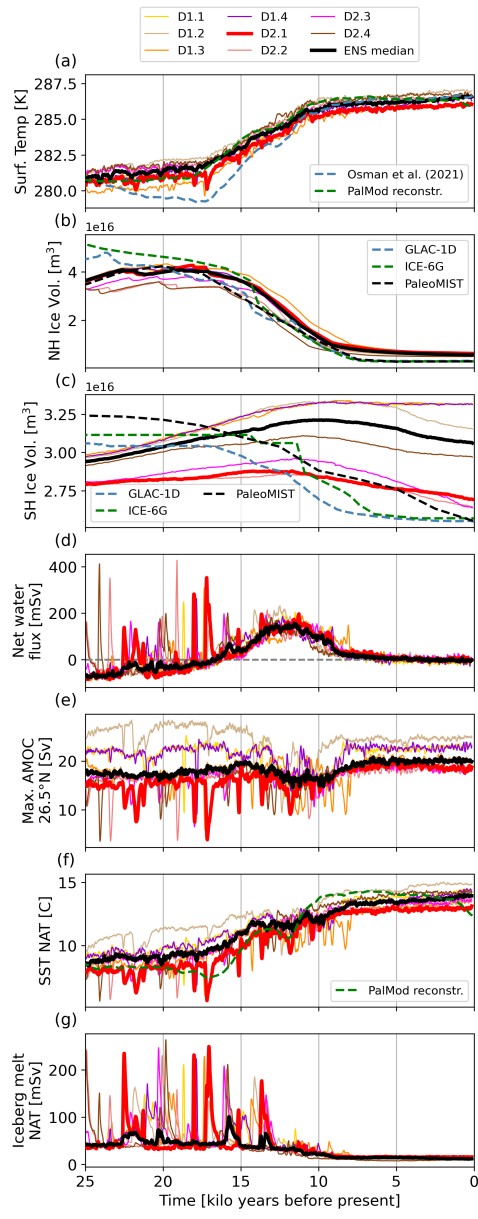

**Figure 8.** Time series of global mean (a) near-surface air temperature, ice volume in the (b) northern hemisphere and (c) Antarctica, (d) net freshwater flux into the ocean, (e) strength of the AMOC at 26.5° N, (f) average SST of the North Atlantic (north of 30° N) and (g) iceberg meltwater flux into the North Atlantic for the model ensemble. The ensemble median is plotted with a thick black line. Model data shown are 100-year running means for climate model variables. Ice volume data are derived from snap shots 50 yr apart. Observational estimates are plotted in the time resolution given by the original data sets. The data from the PalMod reconstructions are anomalies relative to PI. They were transformed into absolute values by adding the last value from the ensemble median.

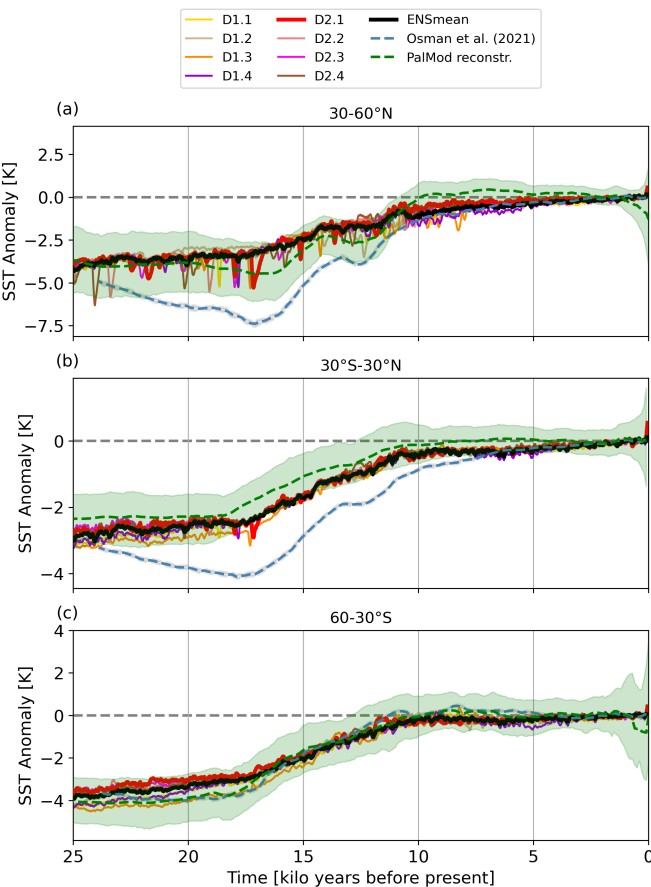

**Figure 9.** Time series of SST anomalies relative to the PI period for different latitude belts: (a) for the northern mid-latitudes, (b) the tropics, and (c) for the southern mid-latitudes. Beside our model ensemble, SSTs from the PalMod reconstructions (green; see Appendix A1) and the reanalysis of Osman et al. (2021, blue) are shown (see also Fig.2). The shading indicates uncertainty estimates, as derived by the PalMod reconstructions. All data were regridded onto a 0.5° grid before averaging over each latitude band. Note, that the data frequency differs: Osman et al. (2021) provide 200-year means of annual SSTs, the PalMod reconstructions provide data with a temporal resolution of 100 years and our ensemble is shown as 100-year running means calculated from decadal mean data.

In the southern hemisphere extratropics, the model simulations agree very well with the reanalysis data by Osman et al. (2021) and lie within the uncertainty range of the PalMod proxy data. All data sets show a slight warming trend through the glacial period. The deglacial warming starts between 18 and 17 ka. The amplitude of the millennial scale variability is relatively small compared to the other latitude bands. After 11 ka the mean SST in this latitude belt remains stable, which is also indicated by the proxy data (Fig. 9c).





*Ice sheets*

At around 25 ka, the simulated northern hemispheric ice volume is lower than the ice volume in the ICE-6G (Peltier et al., 2015) and GLAC-1D (Tarasov et al., 2012; Briggs et al., 2014) reconstructions (Fig. 8b), but matches well with the estimate
from the more recent PaleoMIST (Gowan et al., 2021) reconstruction. The discrepancy in ice volume between the simulated and the ICE-6G and GLAC-1D ice sheets originates from the overall thinner simulated ice sheets at the beginning of the simulation, primarily in the central ice-sheet regions. The maximum ice volume for ICE-6G and GLAC-1D occurs around 25 ka, followed by a steady ice-volume decrease. The same time period in our simulations is characterised by an ice volume gain, as is also shown by the PaleoMIST reconstruction. The long-term mass gain is only interrupted by repeated surge events.
The growth of the ice sheet is directly related to the continuously low greenhouse gas concentrations prior to 21 ka, resulting in rather cold climate conditions (Fig. 8a). In our simulations and the PaleoMIST reconstructions, the simulated maximum ice volume is typically reached at around 18 ka. At this time, the GLAC-1D reconstruction lies also well within our ensemble spread, whereas the ICE-6G reconstruction indicates larger ice volumes than all model realisations. After 18 ka, ice-sheet volume slowly decreases in our simulations, transitioning into the last deglaciation with the strongest ice-volume loss occurring
between 14 and 10 ka. Ice-volume loss tapers off markedly after 8 ka. At PI, our simulations overestimate ice volume in the northern hemisphere by 42-47% in comparison to the reconstructions. The main reason for this is the slow melting of the additional ice caps present in eastern Siberia and over Baffin island (Fig. 7, second column).

As for the northern hemisphere, the simulated ice volume in the southern hemisphere is lower than indicated by the ice sheet reconstructions at the beginning of the simulation at 25 ka (see Fig. 8c). While all reconstruction show either constant ice vol-
ume or a slight decrease after 25 ka, our simulations exhibit an ice-volume gain until 12–10 ka. Most of our model simulations show an ice-volume decrease only afterwards. However, the magnitude of the ice-volume loss is not large enough over the remaining simulation period, resulting in an Antarctic ice sheet that is too extensive and for which the grounding line positions are too advanced for PI. Overall, the simulated and reconstructed ice-volume evolution does not match with reconstructions as well as for the northern hemisphere. Especially, the advance and retreat magnitudes of the simulated Antarctic ice sheet are
underestimated and do not show enough fidelity. As the signals arising from changes in the Antarctic ice sheet are rather small, they did not have an obvious effect on the simulated climate signals during the deglaciation.

*Net freshwater flux into ocean*

The net water flux into the ocean (Fig. 8d), which on centennial mean time scales is almost identical to the time derivative of the mass of the terrestrial ice sheets, aligns with the growth of the northern hemisphere ice sheets during the last part of the
glacial. This growth is interrupted by strong discharge events on millennial time scales. From 17 ka onward, the net water flux remains largely positive as a result of the ongoing decay of the ice sheets, with melt rates (direct meltwater discharge as well as melting ice bergs) larger than 0.1 Sv between 14 and 10 ka. At the end of the deglaciation, around 8 ka, the ice sheet mass loss and hence the net water flux reduces drastically and stays relatively constant through the entire Holocene.





From sea level reconstructions at Barbados, Fairbanks (1989) showed the existence of two meltwater pulses. For the first one,

MWP1A, Deschamps et al. (2012) estimated meltrates larger than 0.46 Sv for more than 300 years. MWP1A coincides with the rather warm period of the BA. Ice-sheet reconstructions such as ICE-6G and GLAC-1D suggest that most of the meltwater was discharged into the North Atlantic or Arctic ocean. None of our coupled simulations produces an amount of meltwater similar in magnitude to MWP1A around 14 ka . However, meltwater pulses close to 0.4 Sv do occur in our simulations during some of the abrupt events. The existence of the second meltwater pulse, MWP1B, before 11 ka was questioned by e.g. Bard

et al. (2010) based on sea level reconstructions from the tropical Pacific and It is not simulated by our model either.

### *AMOC*

The time series of the maximal strength of the AMOC cell at 26.5° N shows marked millennial-scale variability during the glacial period (Fig. 8e). The spread between the ensemble members is considerable. Most simulations show a strengthening of the AMOC from the glacial to the late Holocene. However, there are also simulations with a rather small change in the AMOC

between LGM and PI and one simulation (D1.2) with a stronger AMOC at the LGM than at the PI. During the peak of the deglaciation, all simulations show a weakening of the AMOC, which we explain as the combined response to meltwater hosing and surface warming, both enhancing vertical stability and reducing the ventilation of the deep water masses.

## 4    Abrupt events

The transient climate evolution of the past 25 ka is interrupted by several abrupt events. For each of our ensemble members,

these abrupt events occur at very different times during the glacial and the deglaciation. Here, we focus on the deglacial key events. To examine the drivers of the abrupt events, we first focus on the simulation D2.1. Figure 10b shows the time evolution of the AMOC and the dye fraction over depth at 26.5° N. The overall trend of a strengthening and deepening of the AMOC cell from the glacial to the Holocene is interrupted by numerous abrupt events, which are accompanied by strong signals in the dye fraction and indicate water mass displacements throughout the whole water column (see also Fig. A2). A close relationship

between changes in the net fresh water flux (Fig. 10a) and variations in AMOC and the dye fraction profile is visible.

In the following we will examine individual abrupt events and discuss the underlying mechanisms. Based on our simulations, we can distinguish between three different mechanisms behind the simulated abrupt cold events: HE-like surges from the northern hemispheric ice sheets, and events that are triggered by changes in the Arctic net-freshwater budget or a river rerouting over North America.

### 4.1    Drivers of abrupt climate events during the last deglaciation

### *HE-like surge events*

The first deglacial meltwater peak in D2.1 occurs around 18.0 ka (event 1 in Fig. 10a). The snapshot maps (see Fig. 11a) show that the event at this time is caused by a surge event from the Fennoscandian ice sheet, with massive iceberg discharge into the





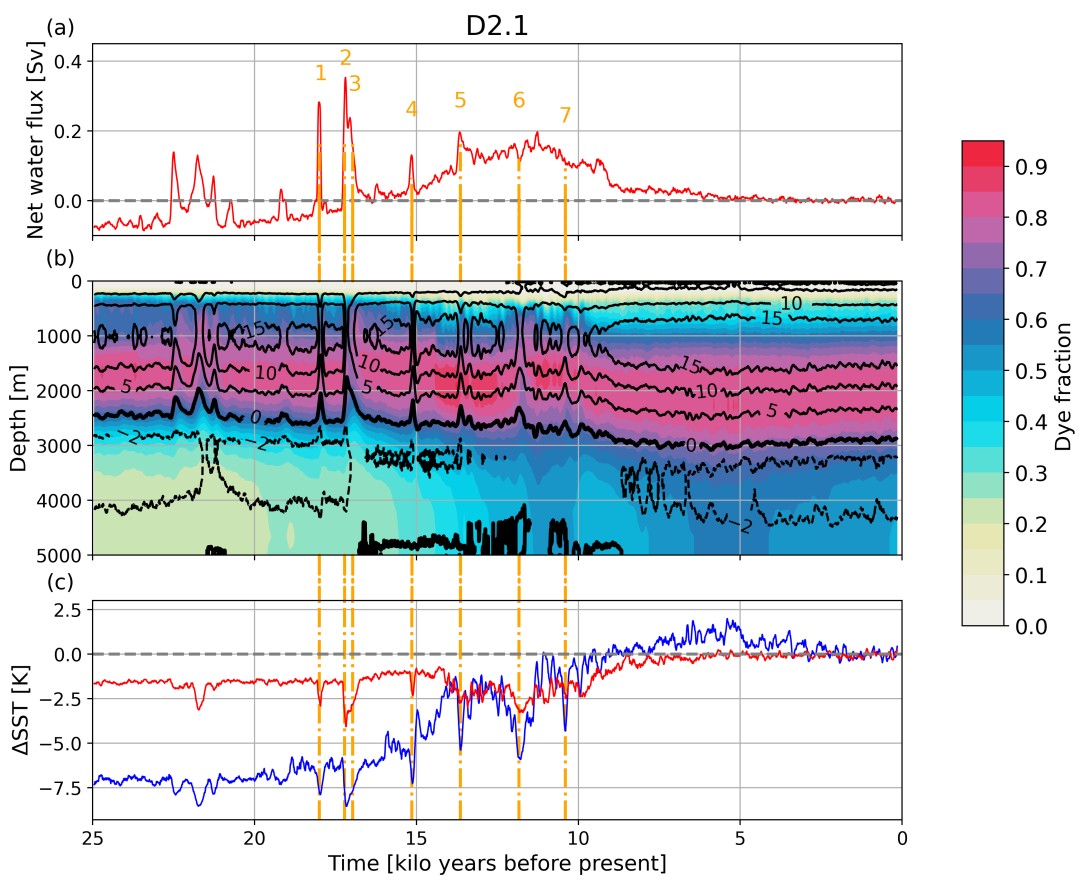

**Figure 10.** (a) Time series of the global freshwater input into the ocean, (b) a time-depth plot of the AMOC at 26.5° N superimposed on the zonally averaged fraction of water originating from the surface of the North Atlantic and Arctic, and (c) SST anomaly time series at 54° N 30° W in the subpolar (blue) and 38° N 46° W in the subtropical (red) North Atlantic for ensemble member D2.1. Orange dashed-dotted lines and orange numbers in (a) mark specific AMOC weakening/ North Atlantic cooling events discussed in the text. Contour lines in (b) are at -2, 0, 5, 10, and 15 Sv. The thicker black line between 2000-3000 m indicates the lower boundary of the NADW cell (AMOC = 0 Sv). Data shown are 100-year running means.





Norwegian Sea. This enhances iceberg melt in the Nordic Seas and the North Atlantic. The meltwater reduces surface salinity,
suppresses deep convection and consequently leads to a weakening of the AMOC and an upward shift of the margin between
AABW and NADW (Fig. 10b). These changes are accompanied by a surface cooling in the North Atlantic (see Figs. 10c and
8f). Note that the SST cooling in the subpolar and the subtropical North Atlantic is of similar magnitude during the glacial and
early deglaciation (Fig. 10c). The glacial temperature at the subpolar location is already rather close to freezing point, which
very effectively limits the potential amplitude of any cooling signal at this location.

The next meltwater peak, occurring around 17.22 ka (marked as 2 and 3 in Fig. 10a), is caused by surges from the Laurentide
ice sheet (Fig. 11b). The event consists of two almost synchronous surges from two source regions of the Laurentide ice sheet.
First, a surge in the western part of the Laurentide towards the Arctic occurs. While this surge slows down, another surge takes
place from the eastern Laurentide through the Hudson Strait (Fig. 11c). This abrupt event is rather strong with a substantial
effect on the AMOC, with a weakening of the AMOC down to 3.9 Sv at 17.2 ka (as indicated in the 100 yr means in Figs. 10b
and 8e). As a result, the margin between the NADW and the AABW cells shifts upward by approx. 800 m. The depth of the
70% NADW dye fraction moves upward and reaches a maximal uplift of about 400 m 200 years later. The two surges combined
discharge a water equivalent of almost 8 m of global sea level within 450 years. This is within the previously reported range
for HEs of 2–20 m over a time period of 500 years (Hemming, 2004; Roche et al., 2004; Roberts et al., 2014) .

The next event, at approximately 15.14ka (marked as 4 in Fig. 10a), is again caused by a discharge event from the Fennoscan-
dian ice sheet into the Norwegian Sea (Fig. 11d). It is followed by the last big iceberg discharge event around 13.64 ka (event
5 in Fig. 10a) from the eastern Laurentide ice sheet through Hudson Strait (see Fig. 11e). As the deglacial warming is already
ongoing during these two events and the temperature at the subpolar location is now well above the freezing point, the SST
cooling signals in the subpolar location are now stronger than in the subtropics, similar to the responses known from typical PI
hosing experiments (e.g., Schiller et al., 1997; Stouffer et al., 2006; Smith and Gregory, 2009).

*Events not caused by ice surges*

After 13.64 ka, the North Atlantic in D2.1 shows two more distinct SST cooling events (marked as 6 and 7 in Fig. 10a). The
first of these events reaches a temperature minimum around 11.83 ka and the other one at 10.4 ka in simulation D2.1 (Fig.
10c). They coincide with a significant decrease in the maximum AMOC strength (Fig. 10b). But, unlike the previous events,
they cannot directly be associated with an abrupt increase of the total amount of meltwater and/or iceberg discharge from the
Laurentide or Fennoscandian ice sheets (Fig. 10a).

*Sign shift in the Arctic net-freshwater balance*

The first of these two cold events takes place during the peak of the deglaciation (event 6 in Fig. 10a). During this time, the
Laurentide and Fennoscandian ice sheets experience significant mass loss in response to the increase in global temperatures and
summer insolation. This is indicated by an increase in global meltwater release (Figure 10a), which is specifically pronounced
in the Arctic (Fig. 12b). This meltwater input into the Arctic leads to a regime shift in the net freshwater budget of the Arctic
ocean. This quantity is defined as the balance between the meteoric freshwater input (river runoff, precipitation and iceberg





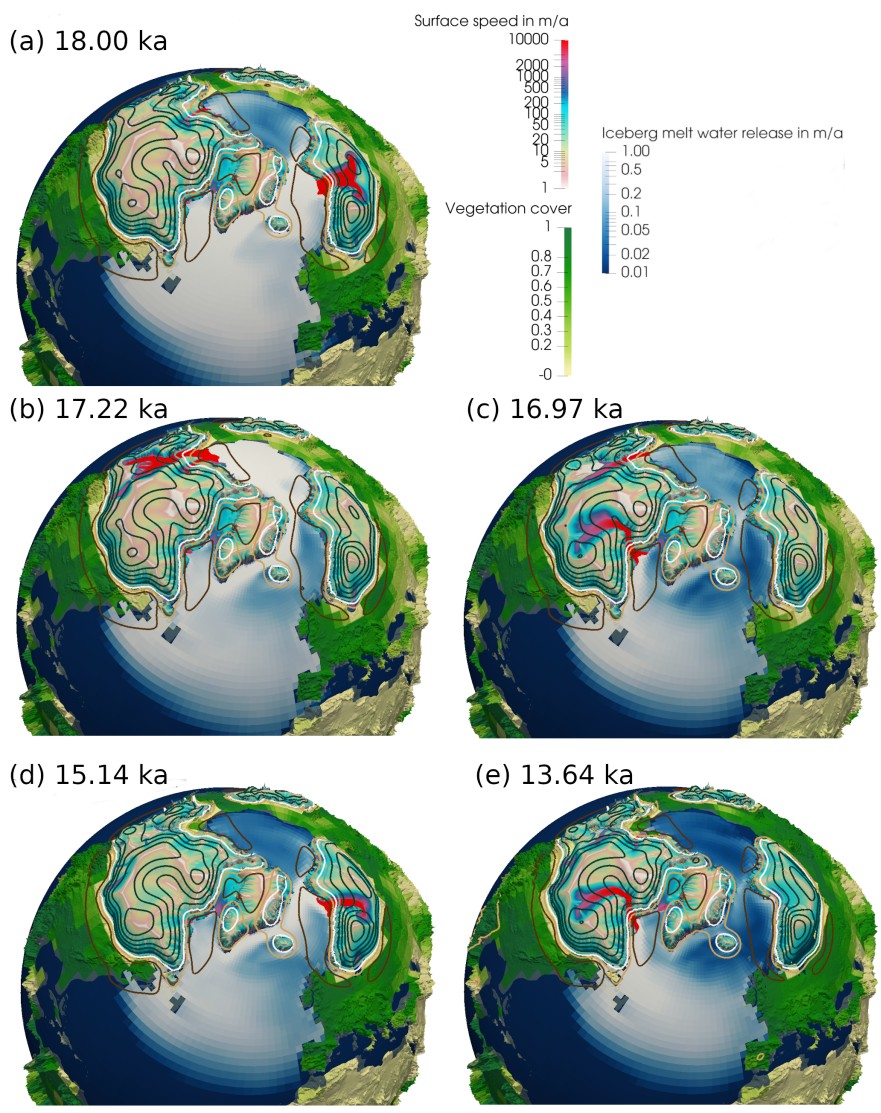

**Figure 11.** Snapshots of different surge events from the northern hemisphere ice sheets during the last deglaciation for simulation D2.1. The movie of the entire deglaciation is available at https://owncloud.gwdg.de/index.php/s/Z1Ce6ZYVN71ffky . Colours on the ice sheets indicate the surface speed, colours on land maximum vegetation cover and colours in the ocean the amount of iceberg meltwater release. Isolines indicate the vertical displacement of the bedrock relative to the end of the simulation. Black isolines indicate depressed bedrock, brown isolines elevated bedrock as compared to PI. No change is indicated by a white isoline. The spacing between individual contours is 100 m with the exception of the ± 50 m contour.





melt minus evaporation) and the export of sea ice from the Arctic into the adjacent seas, and represents a measure for the integrated net freezing rate of Arctic sea ice and the associated brine release. During the glacial, the Arctic sea-ice export and thus the brine release dominates over the meteoric freshwater input. Hence, the water leaving the Arctic has a higher salinity

and colder temperatures and is therefore denser than the inflowing water of Atlantic origin. As a consequence, it contributes to the formation of NADW. Due to the increase in meltwater input from the Laurentide ice sheet into the Arctic ocean, the net-freshwater budget becomes positive at about 12.17 ka. This means that the Arctic transitions from a state with an effective net freshwater loss due to sea-ice formation and export to a state with a positive net freshwater budget (Fig. 12b). The net freshwater input allows for a halocline to develop in the Arctic, which stabilizes the stratification in the Arctic. The water

leaving the Arctic is now fresher and less dense than the inflowing water. This results in a decrease in AMOC strength. While the AMOC weakening is rather gradual at first and overlaid by large centennial-scale variability, a gradual decrease in Arctic sea-ice export starting at about 12.0 ka results in a further increase in the net freshwater budget and ultimately leads to an abrupt weakening of the AMOC (Fig. 12c).

The overall increase in sea level and the retreat of the ice sheets results in the opening of the Bering Strait as well as of

the Barents Sea opening at about 11.37 and 11.18 ka, respectively. In accordance with this the AMOC strength recovers. The opening of Bering Strait leads to a considerable reduction of Arctic sea-ice volume (about 20%; not shown) and sea-ice export to the Nordic Seas (about 20%). Hence, the opening of both straits ultimately increases the surface salinity, as saltier waters from the surrounding oceans can now be imported into the Arctic, decreases the stable stratification, which enhances the deep water formation, and allows for a stabilization of the AMOC.

It is noteworthy that the freshwater input into the Nordic Seas increases rather rapidly around the time of the opening of the Barents Sea Opening at 11.18 ka. This increase is associated with a redirection of meltwater from the southern margin of the Fennoscandian ice sheet. Relatively small changes in the land–sea mask and ice-sheet margin allow for a redirection of these waters from the North Sea towards the northeast into the White Sea and eventually into the Barents Sea, causing an abrupt increase in the freshwater input into the Nordic Seas (see Fig. 12b) and reduction in Nordic Seas surface salinity.

*Changes in river routing*

The second of the non-surge induced cooling events (about 10.4 ka; event 7 in Fig. 10a) is driven by a redirection of meltwater from the Laurentide ice sheet. At this time, much of the meltwater from the Laurentide ice sheet is routed through the St. Lawrence or Mackenzie Rivers and ends its path either in the North Atlantic or Arctic, respectively. The paths of the two rivers follow the southern edge of the Laurentide ice sheet and are separated by a drainage divide located northwest of today's Great

Slave Lake (Fig. 13). At about 10.49 ka, the continued retreat of the northwestern part of the Laurentide ice sheet in conjunction with changes in the land topography through glacial isostatic adjustment, allows for the Mackenzie drainage basin to extend much further southeast, all the way towards the Great Lakes (Fig. 13). This rerouting leads to a redirection of approximately 0.08 Sv of meltwater from St. Lawrence River to the Mackenzie River and results in an abrupt increase in meltwater discharge into the Arctic (Fig. 12b). The freshwater is transported through the coastal boundary currents of the Arctic Ocean towards



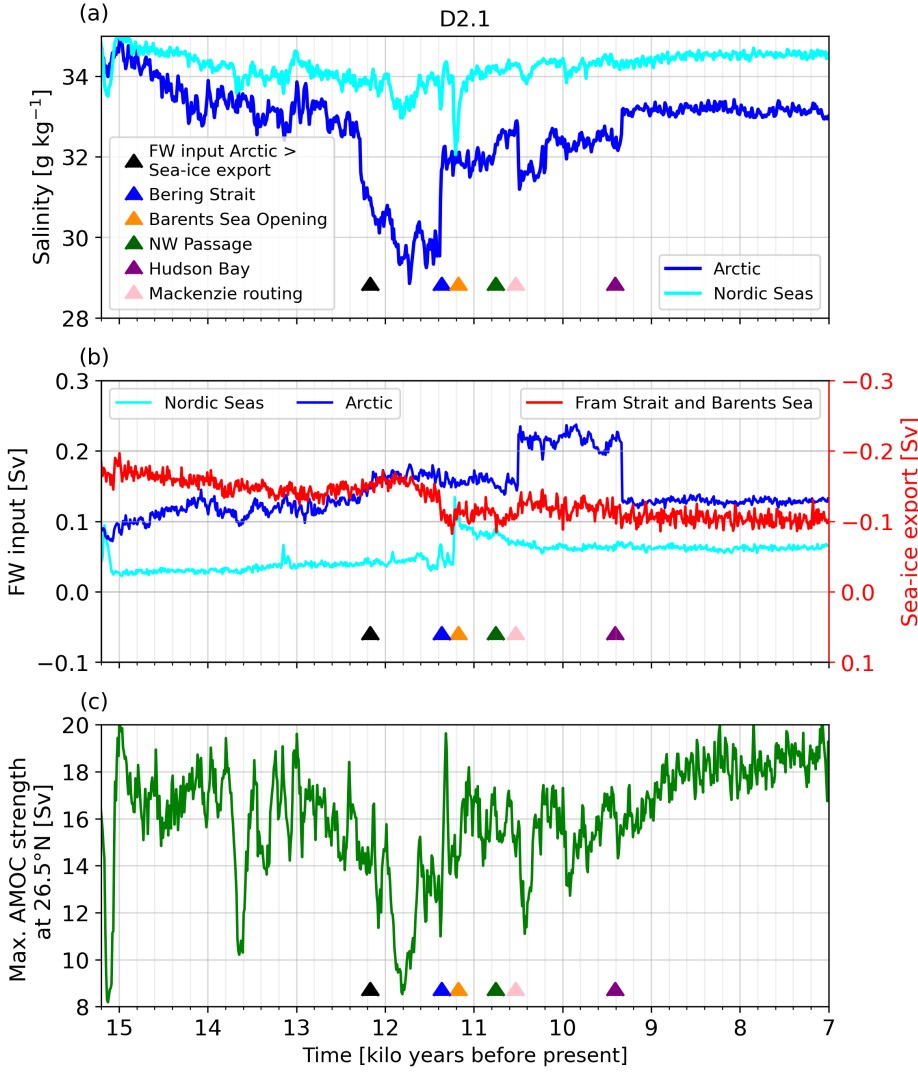

**Figure 12.** Time series of the (a) surface salinity in the Arctic (blue) and Nordic Seas (cyan), (b) the freshwater input into the Arctic (blue) and Nordic Seas (cyan), both derived from precipitation minus evaporation, as well as the Arctic sea-ice export through the Fram Strait and Barents Sea Opening (red; right axis) in simulation D2.1. (c) Time series of the maximum AMOC strength, as presented in Fig. 8d. All data are shown as decadal means. The black triangle indicates when the 100-year running mean of the freshwater input becomes for the first time larger than the sea-ice export, resembling a change in the sign of the net freshwater balance. Other triangles mark the opening of the major Arctic Ocean gateways, namely the Bering Strait (blue), the Barents Sea Opening (orange), the Northwest Passage (green) and the opening of the Hudson Bay (purple). The change in river routing that allows for enhanced freshwater input through the Mackenzie river into the Arctic ocean is indicated in pink.





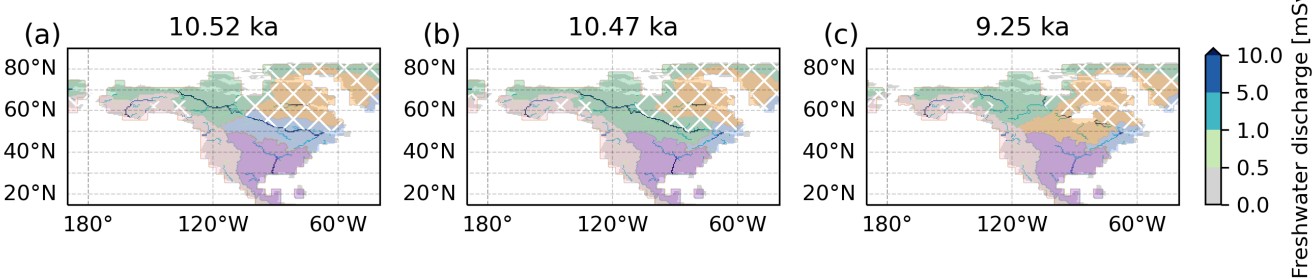

**Figure 13.** Snapshots of the river discharge and their catchments for a time slice before and after the change in the river routing that triggers the 10.4 ka cooling event in D2.1 and after the Hudson Bay opening in the ocean model. Shown is the river discharge through the main rivers (colorbar) over North America at (a) 10.52 ka, (b) 10.47 ka, and (c) 9.25 ka together with the main catchment areas, indicated by the colored overlays. Note that several catchments are combined in one color: shown are all catchments that discharge into the Arctic (green), the Baffin Bay and Labrador Sea (orange), the North Atlantic and Nordic Seas (blue), the southern North Atlantic (purple) and the Pacific (pink). The gray background shading marks the land–sea mask on the ocean model grid. White hatching marks the glacier mask, as obtained from the ice-sheet model.

the deep-water formation sites in the Greenland and Labrador Seas, where it leads to a reduction of the NADW formation and thereby a weakening of the AMOC.

The enhanced discharge of meltwater into the Arctic lasts until about 9.41 ka. At this time, the Laurentide ice sheet collapses and splits into two small ice sheets on Baffin Island and on southeast Canada. The associated changes in sea level and glacial isostatic adjustment lead to the opening of the Hudson Bay (Fig. 12) and allow for a rerouting of water into the Hudson Bay

(Fig. 13). A consequence of the ice-sheet collapse is a short increase in the release of icebergs and their melt into the North Atlantic (Fig. 8g). In response to these changes in the river routing, the AMOC recovers and slowly increases towards its modern strength.

### 4.2 Abrupt events in the ensemble

In this section we discuss, how robust the mechanisms are that we have found in the previous sections for simulation D2.1.

For this we are now discussing the entire ensemble. All ensemble members show abrupt millennial scale events (see Fig. 8) similar to the events presented for D2.1. Abrupt AMOC weakening events lead to a strong cooling in the North Atlantic SST of several Kelvin. Many of these events also show large pulses of iceberg melting in the North Atlantic, indicative of HE-like surge events. Hence, we will distinguish in the following between HE-like surge events, abrupt events induced by a persistent sign shift in the Arctic net-freshwater balance or changes in the river routing.





**Table 2.** Timing of the opening of the Arctic Straits, the change in the sign of the Arctic freshwater balance and the change in the river routing favoring enhanced discharge through Mackenzie river in ka. All events are marked by triangles in Fig. 12. Values for ICE6GP3 and GLAC1DP3 are taken from transient deglacial simulations with prescribed ICE6G and GLAC1D ice-sheet reconstructions, respectively (see Kapsch et al., 2022).

| Name | Bering Strait | Barents Sea Opening | Canadian Archipelago | Hudson Bay | Arctic freshwater | Mackenzie routing |
|---|---|---|---|---|---|---|
| D1.1 | 8.46 | 11.87 | 11.61 | 9.27 | 14.44 | 10.50 |
| D1.2 | 8.91 | 12.15 | 11.93 | 10.86 | 14.62 | 11.11 |
| D1.3 | 7.82 | 10.86 | 10.75 | 8.16 | 13.40 | 10.57 |
| D1.4 | 8.43 | 11.58 | 11.69 | 9.19 | 14.37 | 9.73 |
| D2.1 | 11.37 | 11.18 | 10.76 | 9.41 | 12.17 | 10.51 |
| D2.2 | 16.11 | 12.47 | 12.06 | 9.89 | 13.98 | 10.71 |
| D2.3 | 12.28 | 11.67 | 11.72 | 9.77 | 12.88 | 10.38 |
| D2.4 | 12.69 | 12.75 | 12.96 | 10.84 | 13.62 | 11.73 |
| ICE-6G_P3 [a] | 11.13 | 14.03 | 10.33 | 8.68 | 15.07 | 12.62 |
| GLAC-1D_P3 [a] | 11.19 | 15.08 | 10.43 | 8.44 | 14.98 | 11.56 |
| Proxy evidence | 10.30-13.10 [b] | 16.0-15.0; 12.0 [c] | 8.5-7.5 [d] | 8.6-8.2 [e] | - | - |

[a] Kapsch et al. (2022), [b] Elias et al. (1996); Dyke and Savelle (2001); Keigwin et al. (2006); England and Furze (2008); Jakobsson et al. (2017), [c] Hughes et al. (2015); Rovira-Navarro et al. (2020), [d] England (1999); England et al. (2006) , [e] Tarasov et al. (2012)

### *Abrupt events due to HE-like surges*

HE-like surges are caused by internal variability of the northern hemisphere ice sheets in our model system (Ziemen et al., 2019; Schannwell et al., 2023, 2024), which largely follows the classic binge-purge mechanism suggested by MacAyeal (1993). However, the timing of these events is highly variable as shown by the considerable spread shown between the ensemble members. As the external forcing (orbital parameters and atmospheric greenhouse gas concentrations and for a subset of our ensemble also the volcanic forcing) is identical in all simulations, it cannot serve as explanation for the difference in the timing of the events. As all asynchronous spin-up simulations started from the same state at 45 ka, model parameters obviously have an effect on the timing of the surge events.

To test the role of initial conditions, two additional sensitivity experiments have been performed. Thse simulations are identical to D2.1 except for slight differences in the initial fields for the ice sheet and solid earth model. Instead of the 26 ka conditions, we used the ice sheet and solid earth conditions from 27 ka (D2.1.1) and 28 ka (D2.1.2) in the asynchronous spin-up as initial state. Together with D2.1, the simulations D2.1.1 and D2.1.2 thus allow us to investigate the effect of the initial state of the ice sheet on the timing of the surge events. Of our three main surge regions, Hudson Strait tends to show the most regular interval between individual events, suggesting a higher predictability of surge events than the other regions. D2.1.1 and D2.1.2 simulate Hudson Strait surge events that are offset by 1 and 2-kyrs(Fig. 14a), which matches the shift in





the initial state of the ice sheets and corroborates a high level of predictability shown in previous studies (Schannwell et al., 2023, 2024). In agreement with these studies, the level of predictability is considerably lower for the surges originating from the western Laurentide ice sheet. For this region, the spread in the timing and strength of the events is much more irregular and the simulated shifts in D2.1, D2.1.1 and D2.1.2 are closer to 1 ky instead of the expected 1 and 2 ky (Fig. 14b) from the initialization. The predictability for our third surge region, the Fennoscandian ice sheet, lies somewhere in between the

predictability of the other two surge regions (Fig. 14c). Overall, our sensitivity simulations clearly demonstrate the importance of the initial conditions for the timing of the surge induced abrupt events.

Furthermore, the simulated climate over the ice sheets (especially surface mass balance and surface temperature) has an effect on the timing and frequency of the surge events (Schannwell et al., 2023, 2024). Together with the rather chaotic surge behaviour of the Mackenzie ice stream this further enhances the stochastic contribution in the timing of the surge events. Both,

initial conditions and the choice of the model parameters influence the timing of the surge event. In consequence, the exact timing of surge-type variability, as derived from observations, cannot be reproduced by a model with interactive ice sheets.

### *Late deglacial abrupt events*

The last two North Atlantic cooling events in D2.1, which are not associated with the occurrence of HEs, can be identified also in most other ensemble members (Fig. 8e, f). While the timing of the individual events differs between ensemble members, the

processes leading to abrupt events in D2.1 are similar in all ensemble members.

### Sign shift in the Arctic net-freshwater balance

In most of our simulations, the onset of the first event not induced by ice-sheet surges is associated with the transition into a regime in which the net freshwater input into the Arctic becomes larger than the sea-ice export to the North Atlantic. This leads, similar to D2.1, in all simulations to the formation of a halocline in the Arctic and a significant reduction in the AMOC

strength. However, the threshold of net freshwater input at which the AMOC strength significantly reduces, varies between simulations. For example, D2.2 and D2.3 show weaker AMOC slowdowns and North Atlantic cooling during this onset than the other simulations. This is likely related to the Bering Strait being open already before the event, at 16.11 ka in D2.2 and 12.28 ka in D2.3 (see Table 2), ultimately influencing the freshwater balance and the import of saltier water from the adjacent seas. This is in line with Karami et al. (2021), who showed that the Arctic gateways can have a different impact on the ocean

dynamics in the Arctic, depending on whether the Bering Strait or Canadian Archipelago are closed or open. Our findings are also in agreement with findings by Hu et al. (2012), who showed that abrupt AMOC transitions are more likely when the Bering Strait is closed. In accordance with Hu et al. (2012), we find that the simulations that exhibit a late opening of the Bering Strait show stronger variations in the Arctic ocean and AMOC in response to the shift of the sign in the net freshwater balance than the simulations with an early opening.

In our simulations the Bering Strait also plays a specific role towards the end of the abrupt event initiated by the changes in the freshwater budget of the Arctic. As shown for D2.1, the opening of the Bering Strait leads to a recovery of the AMOC through an increase in salinity of the Arctic and, thus, a decrease in the stability (see section 4.1). This is the case for all but



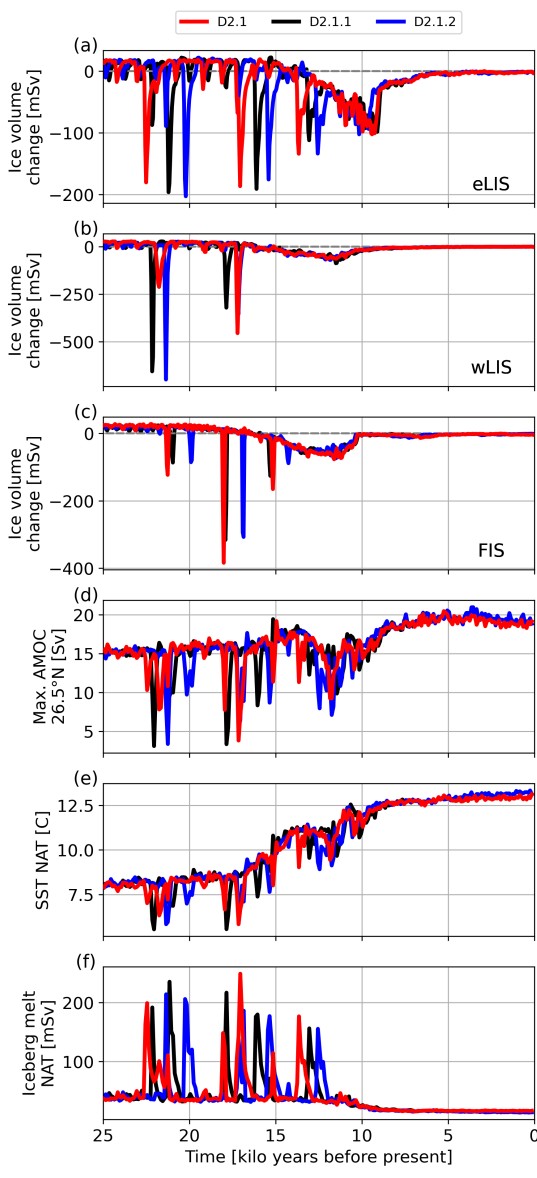

**Figure 14.** Change in ice volume for the (a) eastern Laurentide (eLIS), the (b) western Laurentide (wLIS), and the (c) Fennoscandian (FIS). (d) shows the strength of the AMOC at 26.5° N, (e) the average SST of the North Atlantic, and (f) the iceberg meltwater flux in the North Atlantic for simulations D2.1, D2.1.1 and D2.1.2.





one of the ensemble members. In line with Karami et al. (2021), who found that the opening of the Bering Strait leads to a substantial salinity increase in salinity of the top 50 m of the Arctic ocean, our simulations show a substantial increase in surface
salinity for D2.1 (Fig. 12b) and the majority of our simulations (D1.1, D1.2, D1.3, D1.4, D2.3, D2.4, not shown). However, as the timing of the Bering Strait opening significantly varies between simulations, the full recovery of the salinity also varies. For the simulations with a late opening of the Bering Strait (D1.1, D1.2, D1.3, D1.4), the Arctic is able to maintain low surface salinity and thereby its stable halocline over several millennia. This leads to an overall weaker AMOC with substantially higher variability throughout the course of the deglaciation, until the Bering Strait finally opens (see Table 2).

It is noteworthy that the ocean response to an opening/closing of the Bering Strait differs from other studies in some aspects. For example, Hu et al. (2015) argue that the opening/closing of the Bering Strait has a substantial impact on the sign of the freshwater budget in the Arctic. They find that if the Bering Strait is opened under deglacial conditions (15 ka), the Arctic changes from a freshwater sink to a freshwater source. Our simulations do not show such a relationship. Rather, they show a transition from a typical glacial state, where the sea-ice export from the Arctic to the Atlantic is larger than the freshwater
input into the Arctic, to a state where the freshwater input is larger. Further, our simulations show that this transition is an inherent transient climate response that occurs in all simulations independent of when the Bering Strait is opening. The main difference between the experiments from Hu et al. (2015) and our simulations is that an opening of the Bering Strait in our simulations leads to reduced sea-ice export from the Arctic to the Atlantic, whereas their model simulates an increased export. In our simulations, the reduced ice transport is also a result of a reduced sea-ice volume inside the Arctic.

A similar sign shift of the Arctic freshwater budget and transition into a slightly weaker AMOC state is also evident in simulations with prescribed ice sheets from the ICE-6G (Peltier et al., 2015) and GLAC-1D reconstructions (Tarasov et al., 2012; Kapsch et al., 2022) at about 15.07 ka and 14.98 ka, respectively (see also Table 2). The transition from a state where sea-ice export to the Atlantic is larger than the freshwater input into the Arctic, to a state where the freshwater input dominates is accompanied by the development of a halocline in the Arctic ocean in the simulations with ICE-6G and GLAC-1D recon-
structions (see ICE6G_P3 and GLAC1D_P3 in Table 2). However, the AMOC response in these simulations is much weaker than in the fully coupled simulations (not shown). This is likely related to a relatively low prescribed meltwater input into the Arctic, which was derived from the ice-sheet reconstructions during this time. Only around the time of MWP1A the freshwater input significantly increases. In general, the simulations with coupled ice sheets show an earlier and more gradual increase in freshwater during the peak of the deglaciation, while the freshwater input into the ocean in the simulations with prescribed ice
sheets is dominated by pulses of meltwater. This likely explains the different response.

### *Changes in river routing*

The deglacial North Atlantic cooling event that is associated with changes in the river routing also occurs in all our ensemble members (see Table 2). It is, consistent with D2.1, in all simulations associated with a change in the drainage basin of the Mackenzie River and accompanied by an AMOC slowdown. It agrees with simulations by Kapsch et al. (2022), where ice-
sheets from the ICE-6G reconstructions were prescribed, as well as studies by Tarasov and Peltier (2005) and Condron and Winsor (2012), who indicate that a rerouting of rivers caused the Younger Dryas cooling. Using a high resolution ocean–sea





ice model, Condron and Winsor (2012) show that freshwater discharged from the Mackenzie river is transported through the narrow coastal boundary currents towards the deep-water formation sites in the Greenland and Labrador Seas, where it disrupts the open-ocean convection and thereby weakens the AMOC. In our simulations, the expansion of the Mackenzie drainage

basin, due to a small retreat on the southern edge of the Laurentide ice sheet, leads to an abrupt increase in the freshwater input into the Arctic and, thereby, triggers the simulated AMOC slowdown and accompanying Northern Hemispheric cooling consistent with D2.1. In all simulations the collapse of the Laurentide ice sheet and the opening of the Hudson Bay, which allow for a rerouting of the meltwater previously entering the Arctic ocean to the Hudson Bay (Fig. 13), leads to a recovery of the AMOC. The freshwater reaching the Labrador Sea through Hudson Strait flows southeastward in a coastal current and has

therefore a relatively small effect on the deep convection in the Labrador Sea. Interestingly, the timing of the change in the river routing occurs in some ensemble members before the opening of the Bering Strait (D1.1, D1.2, D1.3, D1.4), hence, in a state where the salinity in the Arctic is already reduced and the stratification is relatively stable. Despite this, all simulations show a distinct AMOC weakening in response to the enhanced Arctic freshwater input, indicating that the process itself is consistent throughout all ensemble members (not shown). The results emphasize that changes in the river routing played a crucial role

in initiating the North Atlantic cooling events that occur in our simulations between about 12 ka and 9 ka (Fig. 8). Further, the AMOC recovery that follows the opening of the Hudson Bay indicates the importance of changes in the land–sea mask in response to the ice-sheet collapse and associated deglacial sea-level rise for the occurrence of abrupt events. As the changes in river routing and the opening of the Hudson Strait are in all simulations related to the gradual retreat of the Laurentide ice sheet, they appear to be deterministic and, hence, seem to be an integral part of the deglaciation.

## 5  Summary and Conclusion

We have applied the newly developed coupled atmosphere–ocean–vegetation–ice sheet–solid earth model MPI-ESM/mPISM/VILMA to the last deglaciation. Topography and river-routing directions are calculated according to the simulated states of ice sheets and solid Earth. By prescribing time varying atmospheric greenhouse gas concentrations, earth orbital parameters and - in most runs - a volcanic forcing as the only time dependent forcings, the coupled model is able to simulate a realistic deglaciation

including abrupt millennial-scale climate events. By performing an ensemble of eight simulations, we show that the simulated climate change between the LGM and PI and the simulated deglaciation is sensitive to the prescribed model parameters, particularly to the vertical background mixing in the ocean. The simulated SST anomaly between the LGM and PI of all simulations is in line with reconstructions based on proxy data. Nearly all ensemble members show a shallower, and at most a slightly weaker, NADW cell during the LGM than during PI. Interestingly, estimating the shallowing of the NADW cell from

changes in the distribution of a proxy-like water mass tracer (artificial dye tracer) indicates a stronger signal than the values derived directly from the simulated AMOC. This demonstrates an additional methodological challenge in estimating past AMOC changes from sediment proxy records.

All ensemble members show abrupt cooling events. For glacial periods and during the early deglaciation, these events are caused by surges from the Laurentide and Fennoscandian ice sheets that result in the release of large amounts of icebergs





into the North Atlantic, the Nordic Seas and the Arctic. The resulting meltwater input into the ocean leads to reduced surface salinity, suppressed formation of deep water and a weakened AMOC. After the end of each surge event, the system recovers to its previous state. This HE-like variability represents a glacial mode of internal climate variability of the coupled model system. However, the timing of the simulated HE-like surge events shows a considerable scatter in the model ensemble. Additional sensitivity experiments confirm, that these events are not entirely determined by the prescribed external forcing but that the

initial conditions - especially of the northern hemispheric ice sheets - as well as the model parameters play an important role for the timing of these events.

The glacial climate of the Arctic in our model system is also characterized by sea-ice export to the Atlantic that exceeds the amount of freshwater input to the Arctic through river runoff, iceberg melting and precipitation minus evaporation. This effectively prevents the development of a stable halocline and causes the formation of water masses that are colder, saltier

and denser than the inflowing water of North Atlantic origin. Throughout the deglaciation, the enhanced meltwater input due to melting ice sheets together with precipitation and river runoff exceeds the amount of freshwater that is leaving the Arctic through sea-ice export. The persistent change towards a positive net freshwater balance leads in all simulations to the formation of an Arctic halocline, which appears in most simulations quite abrupt. This results in the suppression of dense water formation within the Arctic and in a weakening of the AMOC accompanied by a cooling in the North Atlantic realm. How abrupt the

halocline develops thereby depends on the overall state of the Arctic ocean, determined by the in- and export of waters from the adjacent seas through opened ocean passages (specifically the Bering Strait). As the timing of the opening of those passages differs between simulations, the magnitude and abruptness of the AMOC response also differs.

Additionally, river rerouting in connection with the opening of the Hudson Bay can cause abrupt changes in AMOC and North Atlantic climate. These processes become particularly relevant when the size of the Laurentide ice sheet is already

strongly reduced. The chronological order of the opening of the Bering Strait and the river rerouting thereby substantially affects how abrupt the onset and/or offset of the events occur.

In a comprehensive comparison of the glacial and deglacial climate with proxy data, we show that the overall climate trajectory of our ensemble agrees well with observations. The validation of abrupt events in the model with proxy records is not trivial, especially with respect to the timing of the abrupt events. Although it has been shown that HE-like surge events

have a characteristic return period and are to some degree phase locked to the climate changes (Schannwell et al., 2024), the timing of the abrupt cooling events that are associated with ice-sheet surges in our ensemble are non-deterministic and largely depend on the initial conditions and model parameters. The abrupt events associated with an opening of passages or river rerouting are in principle reproducible as they occur in all ensemble members. However, the nonlinearities involved, such as ice-sheet shape, retreat as glacial isostatic adjustment make the simulation of the exact timing of these events sensitive to

model parameters. Consequently, a temporal matching of the simulated abrupt events with the events derived from proxy data is difficult to achieve. In addition, the dating of proxy records is subject to a high degree of uncertainty. In conclusion, the millennial-scale climate variability in models may not align with time scales found in proxy data, so a validation should rather be based on statistical methods, similar to the model simulations of historical climate with El Niño/Southern Oscillation (e.g. Planton et al., 2021), rather than on the individual timing and strength of deglacial key events.





Our novel model framework presented here is a step towards a better understanding of the processes and interactions between different climate components on deglacial time scales. By including for the first time interactive Earth system components, such as ice sheets, icebergs, solid earth, and time-dependent river directions and land–sea mask into a comprehensive ESM, we could reveal plausible and also unexpected chains of processes that explain the climate features observed throughout the last deglaciation. While we mainly focus on the overall deglacial climate response to changes in the external forcing, such as

insolation, greenhouse gases and volcanoes, as well as the abrupt climate events that occur due to the internal variability of the climate system, our ensemble will enable the analysis of a wide variety of processes in each of the individual modeled climate components and between them. Furthermore, it emphasizes the importance of incorporating Earth system components that are traditionally regarded as exhibiting a too-slow rate of change to be relevant for transient climate simulations.

*Code and data availability.* Model data and scripts used for the analysis will be available online on Zenodo. Max Planck Institute Earth

System Model model code is available upon request from the corresponding author. The PISM code for version 0.7.3 is publicly available at Zenodo under https://doi.org/10.5281/zenodo.7541412. The VILMA source code is available upon request from V. Klemann.





# Appendix A

## A1 PalMod reconstructions

The PalMod reconstructions rely on a synthesis of marine proxy records for temperature of the last glacial cycle, the PalMod
130k marine palaeoclimate data synthesis, beta version 2.0.0 (Jonkers et al., 2020). The database includes globally distributed
sea surface temperature proxies based on alkenone, TEX86, long chain diol index, Mg/Ca ratios in planktonic foraminifera, and
microfossil assemblages. The global mean surface temperature is computed based on the selection of records and the algorithm
described in Baudouin et al. (2024). More specifically, the algorithm is an adaptation of the one used in Snyder (2016) to
leverage the potential of the SST record data set fully. At its core, the algorithm aggregates SST records over latitudinal bands
to estimate a 60° S-60° N SST anomaly timeseries. A scaling factor of 1.9, based on an ensemble of LGM simulations (Snyder,
2016), is then applied to derive the global mean temperature. The latitudinal bands and North Atlantic averages are computed
by aggregating the records over the specific domain without applying a scaling factor.

## A2 Effect of vertical mixing on AMOC

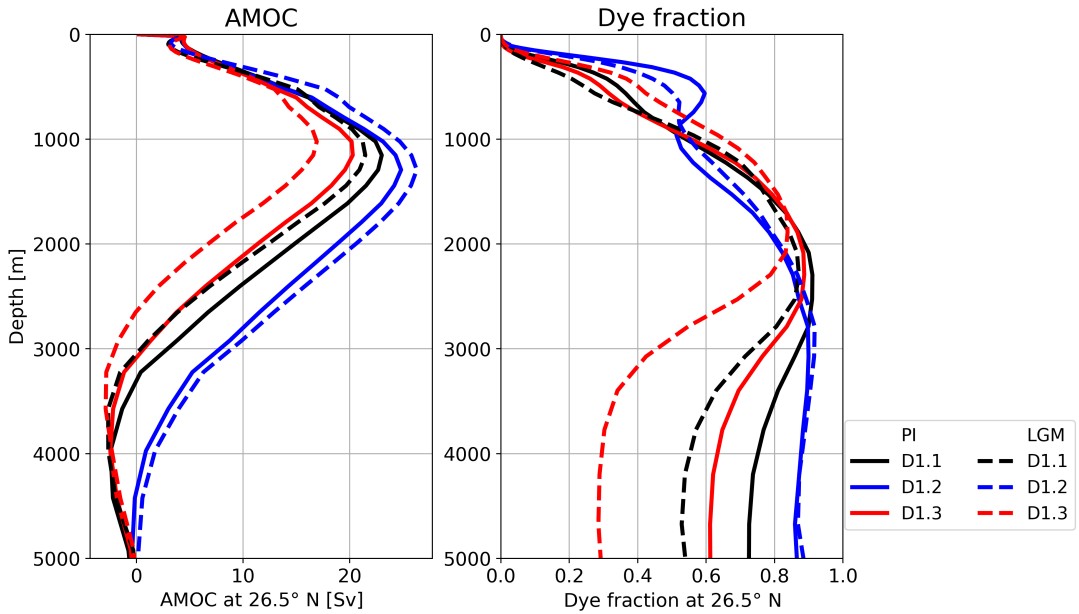

**Figure A1.** As Fig. 4a, c, but for selected ensemble members with different vertical background mixing rates: D1.3 has the lowest rate, D1.2
the highest (see Table A1). Solid lines give the PI state and dashed lines the LGM state.



## A3 Temporal evolution of dye fraction on the lower boundary of NADW cell

Sedimentary proxy data have a long tradition to be used to estimate changes in the AMOC strength and geometry (e.g. Duplessy et al., 1988; Oppo and Lehman, 1993; LeGrand and Wunsch, 1995). In addition to methodological uncertainties intrinsic to sediment proxy data (e.g. biologically induced errors in the recording and preservation of the targeted signal, temporal changes in source functions or measurement errors), the interpretation of the determined variations in terms of changes in AMOC geometry or strength is not straightforward. The dye fraction provides some insights into the temporal signal evolution within a

consistent model system including a well-defined constant source function at the surface (1 for the northern Atlantic and Arctic surface waters and 0 for the remaining ocean surface, including the Southern Ocean deep water source regions).

Figure A2 supplements the information given in Figure 10 and displays the temporal evolution of the zonally averaged dye fraction on the water mass boundary between NADW and southern source deep waters, defined by AMOC=0, for the latitudinal range from 30° S to 40° N over the entire deglaciation. Overall, the dye fraction increases by 0.1 to 0.15 from the LGM to

present day at all latitudes with a larger increase in the southern hemisphere when the NADW reaches further south during PI (Fig. 5). It can be assumed that a similar temporal shift occurs in proxy data, which are used to estimates the AMOC geometry. The interpretation of these proxy data might, in turn, lead to a biased AMOC geometry or to the assumption that the source function of the proxy must have changed over time.

Additionally, we find that the latitudinal pattern in the dye fraction does not evolve uniformly over time. In particular,

larger variations in the dye fraction are present in the southern hemisphere than in the northern during abrupt events. The reconstruction of Atlantic water mass properties from sediment core data which are distributed over the whole Atlantic north of 30° S could be complicated by this latitudinal modulation of the dye fraction. As already stated in the main text, we do not claim that our dye fraction represents real world proxy data, but within our consistent model system we find additional sources of uncertainties which might complicate the reconstruction of the AMOC geometry based on proxy data.

*Author contributions.* UM designed the study and performed all simulations. UM, MLK, CS and KDS analyzed the data and wrote the first draft of the manuscript. UM, MLK, FAZ, CS, MB, OE, VG, VK, VLM, AM, TR, and KDS took part in the development of the coupled model. JPB generated the PalMod proxy reconstructions. All authors critically discussed the results and provided valuable feedback during the manuscript compilation.

*Competing interests.* The authors declare that they have no conflict of interest.

*Acknowledgements.* The project was funded by the German Federal Ministry of Education and Research as a Research for Sustainability Initiative through the PalMod project (grant nos. 01LP1916A, 01LP1917B, 01LP1915C, 01LP2302A, 01LP2316A, 01LP2318A, 01LP1918A and 01L2305A). This work also received funding from the European Union's Horizon 2020 research and innovation programme (Marie



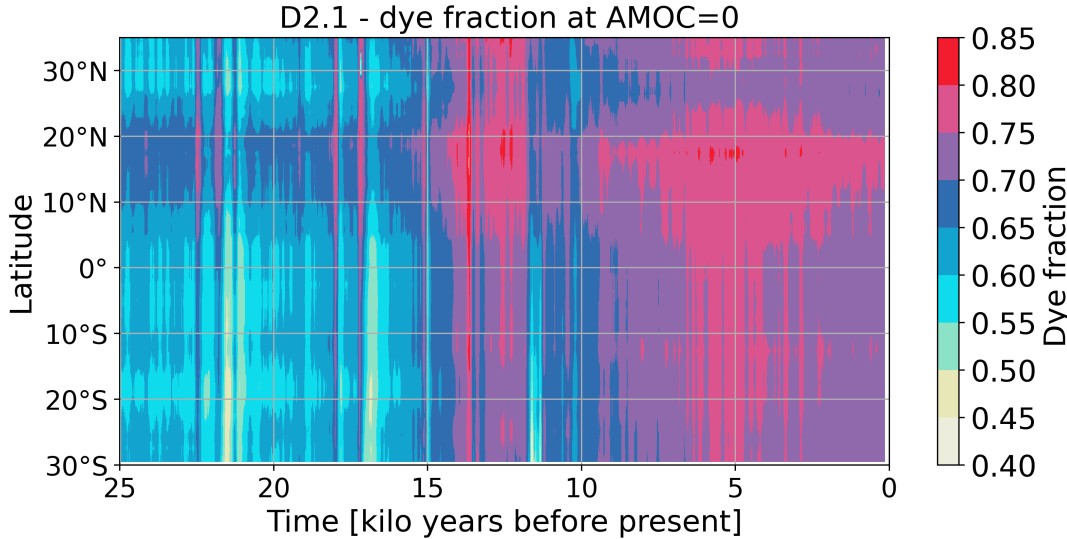

**Figure A2.** Time evolution of the zonal mean dye fraction at the depth of AMOC=0 for latitudes between 30° S and 40° N in simulation D2.1. Data shown are 100-year running means.

Sklodowska-Curie grant agreement No 660893). All model simulations were performed at the German Climate Computing Center. Enrico Degregori optimized the performance of the model code. Matthew Toohey supplied the annually varying volcanic forcing. Thomas Kleinen supported the optimization of the algorithm for the adaptation of the land–sea mask. The authors thank Johann Jungclaus for critical feedback and valuable suggestions.




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



**Table A1.** Individual parameter settings for each model component across the ensemble members and corresponding spin-up simulation.

| Name | MPI-ESM | Volcanoes | PISM_ANT | PISM_NH | Parent run |
|---|---|---|---|---|---|
| D1.1 | version 1 | fixed | version 1 | version 1 | D1.1asy |
| D1.2 | version 1 dback $1.5\times10^{-5}$ | fixed | version 1 | version 1 | D1.1asy |
| D1.3 | version 1 dback $0.7\times10^{-5}$ | fixed | version 1 | version 1 | D1.1asy |
| D1.4 | version 1 | varying | version 1 | version 1 | D1.1asy |
| D2.1 | version 2 ECHAM: crs 0.985 ctr 0.87 cvtfall 2.8 MPIOM: hiccp 27 cw 0.001 dbackprofile 5 dback $4.2\times10^{-6}$ | varying | version 2 sliding_scale_factor_reduces_tauc 1.8 th_heat_coefficient $10^{-2}$ | version 2 sliding_scale_factor_reduces_tauc 1.8 th_heat_coefficient $10^{-2}$ topg_to_phi 8.5, 23.5, -300.0, 400.0 | D2.1asy |
| D2.2 | version 2 atmosphere as in version 1 | varying | version 2 | version 2 | D2.2asy |
| D2.3 | version 2 surface mass balance module: alb_frsnow 0.92 alb_icerefrz 0.55 alb_icemelt 0.55 | varying | version 2 | version 2 | D2.3asy |
| D2.4 | version 2 | varying | version 1 topg_to_phi 14.0, 30.0, -300.0, 400.0 | version 2 | D2.4asy |
| D2.1.1[a,b] | version 2 | varying | version 2 | version 2 | D2.1asy icesheets and solid earth 27 ka |
| D2.1.2[a] | version 2 | varying | version 2 | version 2 | D2.1asy icesheets and solid earth 28 ka |

[a] The sensitivity studies D2.1.1 and D2.1.2 are not part of the ensemble, [b] D.2.1.1 was run only until 7 ka





**Table A2.** List of all parameters and their full name as well as standard value that were varied across the ensemble members.

| Parameter | Explanation | Standard Value |
|---|---|---|
| ECHAM | | |
| crs | relative humidity threshold for cloud formation in the lowest model level | 0.98 |
| ctr | relative humidity threshold for cloud formation in the upper troposphere | 0.85 |
| cvtfall | fall speed of cloud ice | 2.9 |
| MPIOM | | |
| hiccp | ice strenth parameter P* in the Hibler model | 20 |
| cw | drag coefficient between sea ice and water | 0.0045 |
| dbackprofile | ratio between the value near the surface and below 1000 m of the background diffusivity in the ocean | 3 |
| dback | surface background diffusivity | $1.05 \times 10^{-5}$ |
| EBM | | |
| alb_frsnow | albedo of fresh snow | 0.91 |
| alb_icerefrz | albedo of refrozen ice | 0.545 |
| alb_icemelt | melt albedo of ice | 0.54 |
| mPISM | | |
| sliding_scale_factor_reduces_tauc | constant that divides pseudo-plastic tauc (yield stress) by given factor | 1 |
| topg_to_phi | piece-wise linear function of till friction angle | 15, 30, -300, 400.0 |
| th_heat_coefficient | coefficient for turbulent heat exchange from ambient ocean to ice shelf | $4 \times 10^{-5}$ |