# Peer review of "Deglaciation and abrupt events in a coupled comprehensive atmosphere–ocean–ice sheet–solid earth model"

_Climate of the Past, 2024_

## Author Comment (AC1)

Author reply to Louise Sime

We thank Louise Sime for the very helpful comments and suggestions.

The original comments are displayed in red, our replies in black.

Manuscript summary:
This study introduces a novel comprehensive coupled atmosphere-ocean-vegetation-ice sheet-solid earth model, MPI-ESM/mPISM/VILMA, designed to simulate the last deglaciation with high realism. This model is unique because it includes interactive Earth system components such as ice sheets, icebergs, solid earth, and dynamic river directions, allowing for a comprehensive exploration of deglacial processes. The findings highlight that the model can reproduce abrupt millennial-scale climate events and that the timing of these events is influenced by initial conditions and model parameters rather than solely by external forcing. Additionally, the model reveals that changes in Arctic sea-ice export and freshwater dynamics significantly affect the Atlantic Meridional Overturning Circulation (AMOC) and North Atlantic climate.
I find this study really quite exciting. It provides a breakthrough in understanding how different components of the Earth's system interact over long timescales, particularly during the complex process of deglaciation. The ability of the new model to simulate abrupt climate events and reveal unexpected dynamics, including the influence of freshwater dynamics and ice sheet surges on ocean circulation, opens up new avenues for exploring past climate changes and understanding past climate tipping behaviours.
The analysis is mostly well-organised and constructed, and the paper is well-written. It effectively conveys the findings, highlights the novelty of the research, and provides a clear and concise overview of the study's objectives, methods, results, and implications. With the caveats below about it being on the long side, and whether the authors could extend parts of the analysis and split it into two, it is clearly suitable for publication in CP.

Major points:

This is not necessarily a criticism, but I find there is really a lot of material/work in this manuscript. It might be more digestible to most readers if it were split in two, to allow more focus on some aspects. Perhaps with the material on abrupt changes placed into a second manuscript?

We believe that it is an important aspect of this paper to demonstrate that the model's ability to reproduce a large variety of dynamic processes, on glacial-interglacial as well as centennial/millennial scales. Therefore we think it is important to represent them in one paper

The simulations are generally run from 26ka to 1850, allowing a climatological mean PI for each simulation to be specified as the last 1,000 years or so of the run. However, we never see what the PI simulations look like. The manuscript would benefit from an appendix to show how the PI states differ for each version of the model, and some indication of how far they deviate from the observed PI, and a comment on this to be added to 2.2 and 3.1.

Actually, this criticism applies only for SAT. For all other quantities (sea ice, ice sheets and AMOC at 30°N) we show absolute values of D2.1 and the ensemble (median, min and max) as well as observational products in figs. 3, 4 and 7. We will add a Figure to the Appendix that shows the deviations between modelled PI SAT and SAT from the ERA-20C reanalysis (see below). This

Figure also includes modelled and observation based sea-ice extent estimates. We will refer to the figure in the main text.

[Figure]

*Fig.: Difference in annual mean near-surface air temperature between model and ERA-20C reanalysis (colours). Panels (a-h) show individual ensemble members. The isolines show sea-ice extent (> 0.15 sea-ice coverage in the long-term mean seasonal climatology) from the HadiSST dataset for summer (yellow solid) and winter (yellow dashed) and from the model for summer (cyan solid) and winter (cyan dashed). The land-sea mask is indicated by the solid black lines.*

The description of the pre-26ka model spin-up seems to imply that the ocean is spun up for just 10 years, and then the first 1,000 years of each simulation are disregarded. This would give only 1,010 years to spin up the ocean. I think this may not be correct because the text also states that the simulations are initialized from pre-existing glacial simulations. However, this is not very clearly explained. Like the PI point above, spin-up is relatively important, so this should be carefully clarified in 2.1.

This is a misunderstanding. The asynchronous spin-up was run for 19,000 years for ice sheet and solid earth and for 1900 years for MPI-ESM (incl. ocean). We will add some text to make this more clear.

Some clarification/justification for modifications to ocean mixing/vertical diffusion would be useful, both in the context of expected glacial-interglacial ocean mixing changes (due to bathymetric and stratification effects on the dissipation of internal tides/waves). Alongside this, a strengthened discussion of how AMOC and mixing may be affecting the results from this model, given abrupt changes results are relatively strongly dependent on the strength of the AMOC & mixing i.e. discussion of would we expect other GCMs/ESMs to behave similarly (if they also had the same extra component coupled onto them, and were run for similar experiments), or are these unique to this model?

The background mixing is a rather poorly constrained parameter. Therefore we tested in ensemble 1 its effect on the simulation of LGM and the deglaciation. The effect on the AMOC was quite large, as discussed in the paper. The effect on the amplitude of the abrupt events was rather moderate. The size of the ice surges is determined by the ice sheet, only the strength of the reaction of the AMOC on these surges is affected by the ocean mixing and thus the AMOC. The most notable effect was the delay of the simulated deglaciation in case of reduced background mixing as a consequence of the colder polar climate and an early deglaciation in case of stronger mixing. There was, however, a strong negative effect of these changes on the ability of the model to reproduce the observed PI water mass age distribution in the North Pacific. Exp. D1.1 yielded the most realistic distribution in terms of PI radiocarbon (not shown and discussed in this paper, going to be a separate paper). Therefore we modified in ensemble 2 the background mixing only in the upper 1000 m to avoid the strong effects on age distribution. We will add a remark on this in the text.

The section on Abrupt Events is very sensibly laid out; I like the analysis. However, I find it slightly surprising that the focus is solely on simulated abrupt cold events, given there is considerable community interest in the possibility of abrupt warmings too, and some deglacial events are indeed abrupt warmings, rather than solely coolings. If the authors intend to retain the focus solely on abrupt coolings, then it would be helpful to have a sentence or two added to the Introduction and Section to better justify this focus. Otherwise, the focus could perhaps be broadened to also include abrupt warmings, which are also visible in the timeseries provided. See also the first main comment about splitting this manuscript in two.

To keep the paper at a reasonable length, we are focusing on events, where AMOC changes play a major role. By nature, these are mostly abrupt cold events and some abrupt warmings due to AMOC recovery. We will add a remark in the introduction.

Minor points:

The model name is a currently a bit of a mouthful, have the authors also considered giving the model a shorter name too MPI-ESM-extended or similar, perhaps?

The name is a bit long, but there are many model versions around e.g. the version used for the deglacial runs with prescribed ice sheets (Kapsch et al. 2022), a similar version but with interactive methane Kleinen et al. 2020), a deglacial version with carbon cycle and without interactive ice sheets (Extier et al. 2022), and a version with interactive icebergs (Erokhina and Mikolajewicz

2024) plus some older versions with interactive ice sheets (e.g. Ziemen et al. 2019). We believe it is important to make clear, which model version we are using to avoid confusion. The original name is used already in the ongoing cmorization process. Using different model names in the paper and the corresponding data publication is undesirable. Therefore we stick to the name chosen.

L18, missing refs, for very different rates of changes

We will move the citation of Fairbanks (1989) as reference also for the variations to the end of the sentence and add the citation of Lambeck et al (2014) who also estimated time evolution of sea-level rate changes (their figure 4D).

L19-22, greater and lesser 'volumes' rather than changes?

We replaced 'changes' with 'values'.

L34 abrupt 'AMOC' changes, or what events?

L34, 'the quantification of' rather than 'the exact changes'

We will rephrase the sentence in L34 as follows:

These data indicate the existence of abrupt climate events which shaped the sediment record, but even a qualitative estimate of changes in the characteristics of the AMOC remain poorly constrained (e.g. sign of AMOC strength variations....)

L54, and after, clarify what 'CMIP-style' means. Either CMIP models, or perhaps models that use CMIP-atmosphere (AMIP?) models, or perhaps that are run at common CMIP atmosphere-ocean resolutions? Either way, replace 'CMIP-style' descriptions with something more meaningful.

We will not use the term "CMIP-style" anymore, but use "comprehensive climate model" instead plus a more accurate description.

L75, split this into two sentences.

Will be done.

L113, clarify what is meant here by radial directions – by depth, but not by lat or long?

We will change this to: "VILMA is employed in its 1D configuration, which assumes that the viscosity structure of the Earth varies only with depth but is horizontally homogeneous."

Section 2.2 and 3.1, please see above first two main points.

As mentioned above, we will add a plot of the mismatch of PI SAT for all runs including sea ice extents. Ensemble values for sea ice extent and ice sheets are already shown in the existing figures 3 and 7.

Table 1 headings, spell out the headings better. Information about spin-up is not clear. Better to replace the last column 'Parent run' with a much clearer verbal description. Exp file names can be omitted.

We will replace the term 'parent-run' by 'spin-up' and add some more explanations to the table caption. As the individual parameters of the spin-up run are matching the experiments, it would repeat the information already given in column 2. Much of the confusion seems to come from the misunderstanding about the spin-up. We introduced the experiment names of the spin-up more

clearly. Experiment names are important, as all members of ensemble 1 use the same spin-up simulation.

~L125, consider to add a short subsection or para in 2.1 which describes the water tracer/dye methodology, including precise conditions for tracing/dying water, and how tracers/dye is reset (presumably without resetting on exposure, the ocean would eventually saturate?).

We will move the introduction of the dye tracer to the model description and include a figure of the source region in the Appendix. We will emphasise the resetting of the dye tracer to 0 outside of the source region.

L193, maximum?

Will be changed.

L200-202, better to replace this with an Appendix that more carefully shows what the PI states look like themselves (not just the LGM-PI anomalies).

Fig. 3 already shows absolute values of sea ice extent. Both PI values of the model ensemble (Fig. 3a-d) and 'observed' sea ice extents for summer and winter are shown (Fig. 3e+f).

In addition, we will add a new figure to the appendix showing PI SAT biases and sea ice extent for each ensemble member.

Figure 3, obscures more than it shows. It might be more instructive to see the LGM-PI anoms subtracted from the equivalent Anna et al and Osman et al anoms.

We do not share this point of view. For entirely observation-based estimates showing model obs differences might be a good approach. However, the Annan et al. and Osman et al. estimates are strongly influenced by their a priory choice of the models from the PMIP ensemble. So the plots would greatly differ depending on our choice of reference. Therefore we prefer to show the original figures.

Figure 5, I really like these dye/tracer sections.

Thank you

Figure 5 and Figure 6, and thereabouts in text, please ensure there is sufficient information in the text to reassure the reader that these results are not due to spin-up issues. See also main points, and minor point aboves about better clarification on spin-up procedures and ocean run durations.

Prior to the LGM section time slice, the AOGCM was run either 3000 (D1.2 and D1.3) and 4900 years (all other runs) without parameter changes starting from a glacial state. For the LGM climatologies we show the average over model years 3001 to 8000 since the begin of the synchronous simulations. So drift should not be a real problem. PI had about 25,000 synchronous years (+ 1900 asynchronous). In a transient simulation with transient forcing, no full equilibrium can be achieved, see e.g. fig 10b. We do not see any obvious drift in this figure prior to the LGM.

Page 6, and generally dye/tracer results. Please clarify how dye/tracer is removed. Is it reset to 0 on exposure to the atmosphere? Or something else?

We will mention that the dye tracer is set to 0 in the surface ocean outside the source region which guarantees a potential saturation with time (see also reply to L125).

L295, it is interesting that these coupled model simulations do not help much with this Antarctic ice sheet extent problem, possibly it is worth highlighting that this results supports the idea that the problem is in the representation of ice sheet physics in PISM rather than in the forcing/coupling?

We do not believe that this supports the hypothesis that the representation of ice-sheet physics in PISM is insufficient. This is corroborated by the fact that previous studies using PISM for the Antarctic ice sheet over similarly long time scales did manage to reproduce the advance and retreat pattern of the Antarctic ice sheet (e.g. Albrecht et al. 2020, Albrecht et al. 2024). Rather, we think that the parameter space in which our model setup can successfully simulate the advance and retreat pattern of the Antarctic ice sheet could be smaller. We will add a sentence reflecting this to the revised manuscript.

L310-311, rewrite this sentence – it is very hard to understand.

We will reformulate this sentence to: "This variability is not evident in the proxy based products due to the lower temporal resolution of the reconstructions, which is a consequence of the methodological design and the quality of the underlying proxy data."

L375, missing punctuation/sentence issue.

Will be changed.

Table 2, this table would be easier to digest if the rows were shaded to reflect whether the simulated even occurs earlier, later, or (within the uncertainties) at commensurate with the proxy evidence for a similar event.

Thanks for this suggestion. As the timing of the opening of the straits is not consistently earlier or later in individual simulations, a shading reduces the readability of the table. Hence, we decided that we would like to keep it as it is.

L529, salinity twice

Will be changed.

L531, 'varies significantly' rather than the significantly varies

Will be changed.

L581, see first comment on model name

We will keep the model name, see our reply on that above.

L602, remove 'also'?

Will be removed.

L607-610, there are some odd clause orders in here. Check ordering for English, and improve the sentences.

We will rework the mentioned sentences and improve their readability.

L632, either clarify what is meant by unexpected, or possibly rewrite this sentence to focus on the successfully model simulation of hereto uncaptured processes? (river rerouting, arctic freshwater sign changes, strait flow impacts, and other ice-sheet change related climate-land surface-ocean related processes.).

Good suggestion. We will do this.

References not already cited in the paper:

Albrecht, T., Bagge, M., and Klemann, V. (2024). Feedback mechanisms  controlling Antarctic glacial-cycle dynamics simulated with a coupled  ice sheet–solid Earth model, *The Cryosphere*, 18, 4233–4255. doi:/10.5194/tc-18-4233-2024

Extier, T., Six, K., Liu, B., Paulsen, H. & Ilyina, T. (2022). Local oceanic CO2 outgassing triggered by terrestrial carbon fluxes during deglacial flooding. *Climate of the Past*, *18*, 273-292. doi:10.5194/cp-18-273-2022

Kleinen, T., Mikolajewicz, U. & Brovkin, V. (2020). Terrestrial methane emissions from the Last Glacial Maximum to the preindustrial period. *Climate of the Past*, *16*, 575-595. doi:10.5194/cp-16-575-2020

---

## Author Comment (AC2)

**Review by Sam Sherriff-Tadano**

We thank Sam Sherriff-Tadano for the very helpful comments and suggestions.
The original comments are displayed in red, our replies in black.

Summary
Mikolajewicz and others present the ensemble simulations of the last deglaciation with a coupled comprehensive atmosphere–ocean–ice sheet–solid earth model. All the simulations reproduce the general characteristics of climate and AMOC at the LGM and the deglaciation within the uncertainties of reconstructions. The authors further show intriguing results, such as the effects of ocean background mixing on the AMOC, the comparison of the dye tracer with the traditional definition of AMOC depth (AMOC=0 line), the regular and chaotic aspects of ice serge and their impacts on the abrupt climate change and effects of the opening of Bering Strait and the river routine on the abrupt climate changes. Particular attention is also given to the freshwater budget analysis in the Arctic when explaining non-surge-related abrupt climate changes.
The results presented in this study with their complex coupled earth system model are all exciting and are of interests of the readers of Climate of the Past. I also believe that the simulations performed in this study will greatly contribute to advance the science of the climate community. While, I recommend an intermediate revision based on the comments listed below, I do feel these results should be published. Thanks for all the effort in conducting this study!

General comment

My main concern focuses on the analysis of freshwater budget over the Arctic region (Fig. 12) and its effect on the AMOC.

The abrupt AMOC changes in the second half of the deglaciation are explained by the balance of freshwater input and sea ice export in the Arctic, however some ambiguities remain. For example, the rapid salinity drop in the Arctic Ocean (Fig. 12a, blue) precedes the timing of the shift in the net freshwater budget defined by the author (Fig 12 black triangle), though this was not explained. Perhaps, gradual changes in oceanic freshwater transport may affect the exact timing, but please clarify this point.

This is correct. We believe that gradual changes in the transport are the cause for the abrupt salinity drop. Note also, that the black triangle are chosen, where the 100-year running mean of the freshwater input becomes for the first time larger than the sea-ice export. There is substantial variability in this variable, which is also stated in the text *"While the AMOC weakening is rather gradual at first and overlaid by large centennial-scale variability, a gradual decrease in Arctic sea-ice export starting at about 12.0 ka results in a further increase in the net freshwater budget and ultimately leads to an abrupt weakening of the AMOC"*. To visualize that the changes are gradual, we will replace the triangles by a transient line, which indicates the dates for which the unfiltered decadal mean time series of the Arctic freshwater budget crosses zero for the first time and the last time.

[Figure]

Modified Fig. 12

Second, the change in AMOC in response to the opening of Bering Strait is also explained by means of sea ice export. However, the role of changes in oceanic freshwater transport between the Arctic Ocean and the North Pacific is not discussed, which could be important (e.g. Hu et al. 2012). Please add this analysis.

Thanks for this suggestion. We will add a short discussion on the role of the Bering Strait and the North Pacific. For this, we have calculated the freshwater export in form of liquid freshwater and

sea ice through the Bering Strait, using a similar approach as Hu et al. (2008) by taking the global mean salinity as reference salinity to calculate the transport (note, that the global mean salinity is changing over time in our simulations). Similar to Hu et al. (2008), we find that freshwater is exported from the Arctic into the Pacific once the Bering Strait opens (see figure below). This enhanced export likely also contributed to the abrupt increase in salinity of the Arctic ocean that coincides with the Bering Strait opening. We will modify the text accordingly  and add a discussion on Hu et al. (2008, 2012) in respect to the Arctic/Pacific freshwater transport.

[Figure]

*Fig. Mean surface salinity  of the Arctic and the Nordic Seas (a) and calculated freshwater export (liquid + solid) from the Arctic to the North Pacific (b).*

Lastly, I would like to suggest the authors to add a discussion on the stability of the AMOC in this model. In the simulations of deglaciation, the AMOC basically remains stable in its vigorous (or so-called interstadial) mode; e.g. even after a drastic weakening due to substantial freshwater input, the AMOC rapidly recovers to the strong mode once the freshwater flux ceases. This may be influenced by the presence of the North American ice sheet, but if the AMOC remains stable in the strong mode throughout the deglaciation, it may be more challenging to reproduce the  Heinrich Events (HE) and BA-like events. Given that the introduction mentions the relationship between freshwater input and BA, it would be valuable to expand on this point in the discussion or conclusion.

We agree with the reviewer that the mono-stability of the AMOC in our model is a point worth discussing. We will add a remark on the mono-stable AMOC around L396, where we will also mention the mono-stability of other versions of MPI-ESM (Jackson et al. 2023). In addition we will discuss the mono-stability in the conclusion.

HE events are Eigenoscillation of the Laurentide ice sheet leading to occasional ice surges. This is largely independent of the AMOC stability. However, the amplitude of the climate response on these surges clearly can depend on the AMOC stability. We will discuss this in section 4.2.

Minor comments

L43-44: Not an easy sentence to understand. Could you explain a bit more, please?

We will revise the sentence:

"However, the sources of uncertainty in this methodology are numerous and diverse. They include an incomplete understanding of the recording and storing of climate signals through proxy data (Dolman and Laepple, 2018; Liu, 2023), as well as a methodological weakness in the reconstruction of the water mass mixing based on proxy data. "

L60: Please cite Snoll et al. (2024, https://doi.org/10.5194/cp-20-789-2024)

We will do so.

L90: Would be useful to describe the climate sensitivity of the AOGCM somewhere in this subsection.

We did not perform the simulations (pictrl and abrupt4co2) which are required to calculate the climate sensitivity. Beside this, for such short time-scale simulations and analysis annual coupling frequency between mPISM and MPI-ESM would be more appropriate. We consider these numbers to be important for simulations of future climate change, but not so much for the transient simulations shown here.

L119: Could you elaborate a bit more on how the SMB are given to the ice sheet model? Is it a 10 year average?

We will add some more details in our model description.

L131: Just wanted to make sure, but in D1.2 and D1.3, changes in ocean mixing is applied after the spin-up, right? If so, please clarify it wherever appropriate.

It was already clearly mentioned in L 143/144 and shown in table 1. We will additionally mention it in the figure caption of fig. A1.

L149: Could you briefly explain the reason of changing the sea ice parameter, please?

Our original choice of the sea ice strength parameter Pstar was based on Hibler 1979. More recent papers e.g. in Hunke and Dukowicz (1997) or Mehlmann and Korn (2021) use a similar high value. The reduced drag coefficient (cw) between ocean and sea ice is motivated by the fact that the catabatic winds in this model are somewhat underestimated due to the coarse resolution and thus the sea ice transports. We will mention this in the revised paper.

L150: Could you briefly describe the reason why the modifications in background mixing is applied only at the upper ocean, please?

The changes in background mixing had a strong effect on the age distribution in the deep Pacific. In order to avoid these undesired effects in the age distribution but to benefit as much as possible from the effect of the reduced background mixing, we restrict the changes to the upper 1000m. We will add a sentence in the text.

L222: Nice results! This might be a difficult question, but why does this model manage to reproduce the LGM AMOC, but not the model version(MPI-ESM) submitted to Kageyama et al. (2021)? Resolution, meltwater from ice sheet or shape of the ice sheet? Please add a sentence on this point.

The behaviour of the standard MPI-ESM-LR is relatively close to model P1 in Kapsch et al. 2022. Our model D1.1 is rather similar to model P3, except for the iceberg component. We are not discussing the MPI-ESM data submitted to the Kageyama et al. (2021) in this paper. We would have to introduce the model simulations presented in Kapsch et al. (2022), which use prescribed ice sheets, for a detailed discussion. This is outside the scope of the paper, as the LGM AMOC is only a small part of this paper. One of the key aspects for the difference in the AMOC response is probably the reduction of the warm bias in AABW.

L269: While it is displayed in Fig. 8, I think it would be extremely useful to summarize the volume of individual ice sheets at the LGM and PI in a independent Table. This would be a great reference for the readers.

Thanks for the suggestion. We will add a table listing these quantities to the appendix

L271: This is a common problem in other climate-ice sheet model simulations so it might be a good idea to refer to those studies as well (e.g. Ziemen et al. 2014, Sherriff-Tadano et al. 2024, https://doi.org/10.5194/cp-20-1489-2024)

We will add the reference to the papers mentioned.

L360: It's interesting to see the decoupling of Antarctic ice sheet and NH ice sheets, despite the climate model reproducing the temperature evolution of SH high latitudes (FIg. 6c). What happens to ice calving of Antarctica ice sheet in response to sea level rise due to NH ice sheet melting (e.g. Gomez et al. 2020, https://doi.org/10.1038/s41586-020-2916-2)?

The coupling between the northern hemispheric ice sheets and the Antarctic ice sheet remains unchanged during the simulation. Evidence for this is that the Antarctic ice sheet does respond to sea-level changes during Heinrich Events. However these changes are short-lived and unlike in the study of Gomez et al., they do not seem to cause the Antarctic ice sheet to cross any stability threshold. Having said this, the ice volume trajectories between northern hemispheric ice sheets and the Antarctic ice sheet diverge in most simulations deeper into the deglaciation. We think that this is primarily caused by different processes driving ice-sheet retreat in the respective hemispheres. In the north, ice sheets are predominately land-based and their decay is driven by atmospheric processes. In the south, large parts of the ice sheet are marine-based and here basal melting regulates most of the simulated retreat.

L382: The effect of surface warming on the AMOC can be state dependent. For example, Zhu et al. (2015, https://doi.org/10.1007/s00382-014-2165-x) showed that surface warming from a glacial state can cause an intensification of the AMOC by reducing the amount of sea ice.

We will formulate the text a bit more cautiously to avoid this problem. We will restrict the effect of the AMOC weakening to the meltwater injection. We will only mention that the slow surface warming enhances vertical stability.

L434-443 & Fig. 12: The abrupt reduction of Arctic salinity (Fig. 12a) precedes the shift in net-FW input over the Arctic (black triangle) and the abrupt AMOC weakening (Fig. 12c), which may imply a role from other processes in reducing the surface salinity over the Arctic and in weakening the AMOC. Could you add an explanation on this point please?

As pointed out in an earlier comment, the shift in the net-FW balance of the Arctic is gradual and cannot be related to one single point in time. To emphasize this point and to visualize that the changes are gradual, we will replace the triangles by a transient line, which indicates the dates for which the unfiltered time series of the Arctic freshwater budget crosses zero for the first time and the last time. This marker indicates, that the shift in the net-FW budget of the Arctic occurs before the abrupt change in salinity.

L441-443: I'm not entirely convinced by this sentence because the role of sea ice export on the AMOC can be quite complicated. For instance, reduced sea ice export from the Arctic can lead to a freshening of the Arctic Ocean, which may increase the oceanic freshwater transport to the Atlantic and weaken the AMOC. Conversely, reduced sea ice export can also decrease sea ice melting in the North Atlantic, potentially increasing salinity in the DWF region and intensifying the AMOC. Therefore, a more comprehensive analysis of the freshwater budget is necessary to clarify this point.

We agree that the dependence of AMOC changes on sea ice export can be quite complex and depends probably strongly on the patterns of deepwater formation. In this particular case, however, the decreased sea ice export leads to the development of a sign shift of the Arctic surface freshwater budget (P-E + river runoff + iceberg melt – sea ice export) and allows the development of a halocline and suppresses deepwater formation in the Arctic. The resulting reorganisation of the overturning circulation results in a weaker AMOC. We will try to make this clearer in the text.

L444-449: I think this paragraph is missing the perspective of freshwater export from Arctic to North Pacific (e.g. Hu et al. 2008 https://doi.org/10.1175/2007JCLI1985.1). Please rewrite the paragraph including this point.

This is correct. We will add a short discussion on the role of the Bering Strait for the freshwater export into the North Pacific (see earlier comment). Similar to Hu et al. (2008), we find that freshwater is exported from the Arctic into the Pacific once the Bering Strait opens. This enhanced export likely also contributed to the abrupt increase in salinity of the Arctic ocean that coincides with the Bering Strait opening. We will modify the following sentences to "The opening of Bering Strait leads to a considerable reduction of Arctic sea-ice volume (about 20%; not shown) and sea-ice export to the Nordic Seas (about 20%) and allows for the export of freshwater through the Bering Strait into the Pacific (not shown)." and will add a discussion on the Hu et al. (2008) in respect to the freshwater transport.

L538: I think it would be helpful to clarify that Hu et al. (2015) drew their conclusion based on simulations without any ice sheet melting flux involved, while this study included the effect of ice

sheet melting in a realists framework. The finding of Hu et al. (2015) was useful to understand the effect of Bering Strait on the net fw input over the Arctic, however as the authors state, the inclusion of ice sheet melting is important in controlling the timing of the shift of the net Fw input over the Arctic during the deglaciation.

Thanks for pointing this out. We agree with this comment and will add "These differences are likely related to Hu et al. (2015) not considering the effect of melting ice sheets, hence, not accounting for the inflow of freshwater from the ice sheets into the ocean."

Fig.3: Sorry if I'm wrong, but I feel the temperature anomaly and sea ice edges over the North Atlantic look inconsistent in (C) and (D). For example, a larger sea ice expansion is observed over the North Atlantic in (C) than (D), whereas the magnitude of surface cooling is larger in (D) than (C) in the same region. By any chance, the temperature anomalies are mistakenly swop between (C) and (D)?

This is a misunderstanding in how panels C and D should be interpreted. Both the temperature anomaly fields and sea ice extensions in C and D show the ensemble max and min at each grid point. So they should not be interpreted as dynamical fields but rather as indicators of the range of variations occurring in our ensemble. Even for temperature, the simulation with the warmest temperature at one place is not necessarily the one with warmest temperatures all another places.

We will try to make this more clear in the caption of Fig. 3.

Fig.A1: In terms of Dye Tracer and AMOC in PI, experiment D1.2 seems the best, but that is the one which does not simulate shallower and weaker AMOC at the LGM. Do you have any comment on that?

Fig. A1 compares the vertical profiles of the AMOC and the dye fraction for different vertical background mixing rates. We are not aware of any robust metric to rate the quality of the 3 experiments. Quantitatively, estimates from proxy data rather indicate a reduction of the dye below 2000m for the LGM compared to PI, which is not simulated in D1.2.

Fig. 7: Please expand the area of the NH so that the readers can compare the full extent of the simulated North American ice sheet with reconstructions.

We will do so.

Fig. 12b: Does the blue line include the input from ice sheet melting and river run-off? It seems so from the main text, but it is not explained in the caption.

Yes, we will make this more clear in the caption.

References not given in the original paper:

Hunke, E. C., and Dukowicz, J. K. (1997). An Elastic–Viscous–Plastic Model for Sea Ice Dynamics. *J. Phys. Oceanogr.*, **27**, 1849–1867.
doi: 10.1175/1520-0485(1997)027<1849:AEVPMF>2.0.CO;2 .

Jackson, L., Alastrué de Asenjo, E., Bellomo, K., Danabasoglu, G., Haak, H., Hu, A., Jungclaus, J., Lee, W., Meccia, V., Saenko, O., Shao, A. and Swingedouw, D. (2023). Understanding AMOC stability: the North Atlantic Hosing Model Intercomparison Project. Geoscientific Model Development, 16, 1975-1995. doi:10.5194/gmd-16-1975-2023

Mehlmann, C., and Korn, P. (2021). Sea-ice dynamics on triangular grids. Journal of Computational Physics, 428, 110086. doi:10.1016/j.jcp.2020.110086

---

## Author Response (AR1)

Dear Editor,

We have modified our ms. as announced in the replies to the reviewers.

Sincerely Yours

Uwe Mikolajewicz and Coauthors

---

## Referee Report (RR1)

The authors have adequately addressed my comments and concerns raised in my first review. Specifically, the discussion around the timing of net freshwater input over the Arctic, opening of the Bering Strait, changes in Arctic sea ice export and salinity and changes in the AMOC are nicely improved. I am very happy to recommend this paper to be published in Climate of the Past. Thank you for all the effort put into conducting this study.

Sam Sherriff-Tadano

---

## Author Response (AR2)

Dear editor,

We have addressed the following remaining comments of the reviewer:
Line numbers refer to the current/updated version of the ms.

1. Clarification and Justification for Ocean Mixing/Vertical Diffusion Modifications: The explanation for modifications to ocean mixing and vertical diffusion remains insufficiently clarified. Numerous studies indicate that significant changes in ocean mixing are to be expected between glacial and interglacial states due to variations in bathymetry (sea level changes), particularly the exposure of continental shelves, and major changes in ocean stratification. These factors significantly influence the dissipation of internal tides and waves, and thus ocean mixing. This needs to be clearly stated and supported with appropriate references. Additionally, this information should be directly linked to the relevant results (e.g., ocean mixing-tuned outcomes) presented in the paper.

We have modified in line 158

"Especially the background mixing is rather poorly constrained."

to

"Schmittner et al. (2015) used a combination of a tidal model and an EMIC to investigate the effect of enhanced tides during the glacial and report a change of global mean diapycnal mixing of more than a factor of 3 compared to PI. Since the background mixing is rather poorly constrained and includes in our mixing scheme a variety of processes, e.g. tidal mixing, we investigated the sensitivity to this parameter in the first sub-ensemble."

In addition we have added at line 261:
"Wunsch (2003) has suggested that the reduced shelf area during LGM lead to stronger tides. Wilmes et al. (2019) applied a tidal model and reports an enhanced energy supply from tides to the internal wave field (1.8 to 3 times higher for LGM than at present, depending on the ice sheet geometries). As we used a time-constant background mixing in our simulations, this effect is not included."

2. Table 1 - Clarity of Last Column: The information in the last column of Table 1 still appears very obscure. Could you replace the entries in this column with meaningful, human-readable descriptions? Currently, they seem more like file names, which detracts from clarity.

They referred to the synchronous experiments, adding "asy" to make clear that this is the asynchronous spin-up.
We have replaced the condensed, but in our opinion very clear last column of table 1 by a human-readable yet somewhat longish text like e.g. 'parameters as in synchronous simulation D1.1'. Table A1 was adapted accordingly. These changes have the disadvantage that the table does not fit well on a page anymore...

3. Table 2 - Improved Readability Through Shading: The readability of Table 2 could be greatly enhanced by shading the rows to indicate whether the simulated events occur earlier, later, or (within uncertainties) in alignment with the proxy evidence for similar events. This is the primary message the table aims to convey. Applying light shading (e.g., light red or blue for later or earlier events, respectively, and keeping rows unshaded if within uncertainties) would make the table much

easier to interpret without compromising its readability. For a similar example of how shading can improve table clarity, please refer to
https://agupubs.onlinelibrary.wiley.com/doi/full/10.1029/2022PA004600.

We did that, but as already pointed out in our previous rebuttal, we do not consider it as an improvement.

Best regards
Uwe Mikolajewicz on behalf of the coauthors

---

## Author Response (AR3)

Dear editor,

in the section data availability we had to introduce temporary links.
The application for permanent dois has been submitted, but we have not received them yet. We will need to update the links during type-setting.

Sincerely Yours

Uwe Mikolajewicz